# CRISPR/Cas9 interrogation of the mouse *Pcdhg* gene cluster reveals a crucial isoform-specific role for *Pcdhgc4*

Andrew M. Garrett[1,2]*, Peter J. Bosch[3], David M. Steffen[3], Leah C. Fuller[3], Charles G. Marcucci[3], Alexis A. Koch[1], Preeti Bais[2], Joshua A. Weiner[3]*, Robert W. Burgess[2]*

**1** Department of Pharmacology and Department of Ophthalmology, Visual, and Anatomical Sciences, Wayne State University, Detroit, Michigan, United States of America, **2** The Jackson Laboratory, Bar Harbor, Maine, United States of America, **3** Department of Biology and Iowa Neuroscience Institute, University of Iowa, Iowa City, Iowa, United States of America

* andrew.garrett@wayne.edu (AMG); joshua-weiner@uiowa.edu (JAW); robert.burgess@jax.org (RWB)

**Data Availability Statement:** All Illumina sequencing data is available at the Sequence Read Archive (SRA) with the accession number

## Abstract

The mammalian *Pcdhg* gene cluster encodes a family of 22 cell adhesion molecules, the gamma-Protocadherins (γ-Pcdhs), critical for neuronal survival and neural circuit formation. The extent to which isoform diversity–a γ-Pcdh hallmark–is required for their functions remains unclear. We used a CRISPR/Cas9 approach to reduce isoform diversity, targeting each *Pcdhg* variable exon with pooled sgRNAs to generate an allelic series of 26 mouse lines with 1 to 21 isoforms disrupted *via* discrete indels at guide sites and/or larger deletions/rearrangements. Analysis of 5 mutant lines indicates that postnatal viability and neuronal survival do not require isoform diversity. Surprisingly, given reports that it might not independently engage in *trans*-interactions, we find that γC4, encoded by *Pcdhgc4*, is the only critical isoform. Because the human orthologue is the only *PCDHG* gene constrained in humans, our results indicate a conserved γC4 function that likely involves distinct molecular mechanisms.

## Author summary

The γ-Protocadherins (γ-Pcdhs) are a family of 22 molecules that serve many crucial functions during neural development. They can combine to form multimers at the cell surface, such that each combination specifically recognizes the same combination at the surface of other cells. In this way, 22 molecules can generate thousands of distinct recognition complexes. To test the extent to which molecular diversity is required for the γ-Pcdhs to serve their many functions, we used CRISPR/Cas9 gene editing to make a series of mouse mutants in which different combinations of the γ-Pcdhs are disrupted. We report 25 new mouse lines with between 1 and 21 intact members of the γ-Pcdh family. Further, we found that for the critical function of neuronal survival–and consequently the survival of the animal–the molecular diversity was not essential. Rather, a single member of the family called γC4 was the only one necessary or sufficient for this function; databases of

PRJNA562238. Analysis outputs are at Mendeley with DOI: 10.17632/yhn7dpmv6v.1.

**Funding:** This work was supported by NIH Grant NS090030 to R.W.B. and J.A.W. and NIH Grant NS055272 to J.A.W. The Scientific Services at The Jackson Laboratory are supported by NIH Grant CA034196. The funders had no role in study design, data collection and analysis, decision to publish, or preparation of the manuscript.

**Competing interests:** The authors have declared that no competing interests exist.

human genome sequences suggest that this important role is conserved. These new strains will be invaluable for disentangling the role of molecular diversity in the γ-Pcdhs' functions, and as we have already found, will help identify specific functions for specific γ-Pcdh family members.

## Introduction

Cell-cell recognition *via* transmembrane cell adhesion molecules is essential for neural circuit formation. With trillions of exquisitely specific synapses in the human brain, it has been suggested that molecular diversity–achieved either *via* alternative gene splicing or combinatorial expression of large adhesion molecule families–plays an important role [1]. In mammals, this is exemplified by the clustered protocadherins (cPcdhs). The cPcdhs, expressed broadly throughout the developing central nervous system (CNS), are cadherin superfamily molecules that engage in strictly homophilic *trans*-interactions. The cPcdh isoforms are encoded by three gene clusters arrayed in tandem at human chromosome 5q31 (encoding 53 cPcdh proteins) and on mouse chromosome 18 (encoding 58 cPcdh proteins)[2, 3]. In mouse, the *Pcdha* cluster encodes 14 α-Pcdhs, the *Pcdhb* cluster encodes 22 β-Pcdhs, and the *Pcdhg* cluster encodes 22 γ-Pcdhs (Fig 1A). While all three *Pcdh* gene clusters contribute to neural development to some extent [4–10], the *Pcdhg* locus is the only one required for postnatal viability, and disruption of this locus results in the strongest phenotypes (reviewed in [11]).

The *Pcdhg* cluster is comprised of 22 variable (V) exons, divided according to sequence homology into A, B, and C subgroups, and three constant exons. Each V exon is regulated by an individual promoter and, upon transcription, is spliced to the constant exons [12, 13](Fig 1B). Each V exon encodes one γ-Pcdh isoform's extracellular domain, including 6 cadherin (EC) domains, a transmembrane domain, and a membrane-proximal variable cytoplasmic domain (VCD), while the constant exons encode a C-terminal cytoplasmic domain common to all 22 γ-Pcdh isoforms (Fig 1B). Single-cell RT-PCR from cerebellar Purkinje neurons indicated that each neuron expressed all three γC isoforms (*Pcdhgc3*, *Pcdhgc4*, and *Pcdhgc5*) from both alleles while stochastically expressing ~4 of the 19 γA and γB isoforms monoallelically [14]. Interestingly, this does not hold in all neurons: single-cell transcriptomics performed on serotonergic neurons indicated that αC2 was the dominant cPcdh, many neurons expressed no *Pcdhg* genes, few if any expressed γC3 or γC5, and some expressed only γC4 [4]. Additionally, few single olfactory sensory neurons were found to express any of the C-type *Pcdha* or *Pcdhg* isoforms [5].

Clustered Pcdh isoforms interact strictly homophilically in *trans*, engaging in anti-parallel interactions involving EC1-EC4, while EC5 and EC6 mediate promiscuous *cis* dimer formation between γ isoforms, as well as with α- and β-Pcdhs [15–19]. These two types of interactions result in a multimeric lattice of dimers between cell membranes sharing the same isoform composition [20, 21], and indeed, homophilic specificity is observed at the multimer level [18, 19]. In this way, the 58 cPcdh isoforms generate thousands of distinct recognition signals. This molecular diversity is paralleled in *Drosophila* by the gene *Dscam1*, which generates 38,016 distinct protein isoforms through alternative splicing [22]. *Dscam1* isoform diversity is essential for neurodevelopmental processes including axon guidance, synapse specificity, and neurite self-avoidance [23–25](reviewed in [26]). Mammalian Dscams, despite important roles in neurodevelopment, do not generate such isoform diversity [27–30].

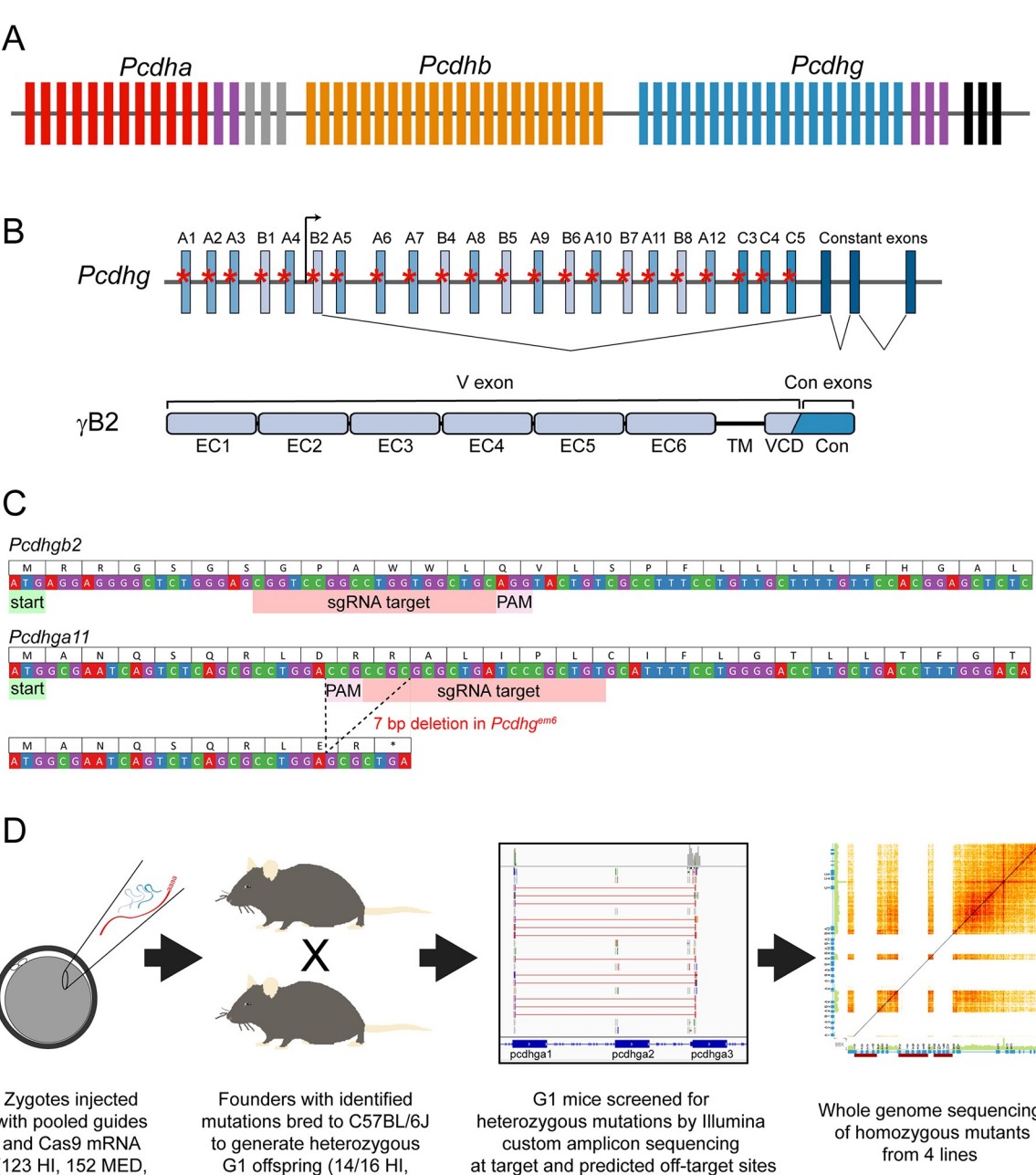

**Fig 1. CRISPR/Cas9 strategy to reduce *Pcdhg* isoform diversity. A**) A schematic of the clustered Pcdh loci with *Pcdha*, *Pcdhb* and *Pcdhg* arrayed in tandem. Constant exons for *Pcdha* are in gray and those for *Pcdhg* in black. The homologous C variable (V) exons are indicated by purple boxes. **B**) The *Pcdhg* cluster is comprised of 22 V exons and 3 constant exons. The V exons are subdivided into γA, γB and γC groups based on sequence homology. sgRNA were designed to target each V exon (red asterisks). As shown for *Pcdhgb2* as an example, each V exon has its own promoter, and upon transcription, is spliced to the three constant exons. In the resulting protein, the extracellular domain (cadherin repeats EC1-EC6), the transmembrane domain (TM) and variable cytoplasmic domain (VCD) are encoded by the single V exon, while the C-terminal constant domain (Con) is encoded by the three constant exons. **C**) Examples for *Pcdhgb2* and *Pcdhga11* indicate the general design strategy for sgRNA, targeting downstream of and proximal to the start codon. An example mutation in *Pcdhga11* (generated in the *Pcdhg^{em6}* allele) leads to a premature stop codon. **D**) The workflow for generating an allelic series of mutants with reduced *Pcdhg* isoform diversity. Values indicate the number of mice–total or from each sgRNA injection concentration–screened at each step.

While α, β and γ isoforms can all contribute to multimer formation [21], the γ-Pcdhs are particularly critical for neural development [11]. Mice lacking either the entire *Pcdhg* cluster or the γC3-C5 V exons exhibited neonatal lethality, with excessive apoptosis of neuronal subtypes in the spinal cord and hypothalamus [31–34]. *Pcdhg* mutants also exhibited reduced synapse number and disorganized synaptic terminals within the spinal cord, which were separable from the increased cell death [35–37]. When neonatal lethality was circumvented with a conditional *Pcdhg* allele, distinct phenotypes were observed in other parts of the CNS, including reduced dendrite arborization in forebrain neurons [38–41] accompanied by an increase in morphologically immature dendritic spines [42]. In retina-restricted mutants, many neuronal subtypes exhibited excessive cell death without separable synaptic disorganization [10, 43]. Starburst amacrine cells (SACs), however, survived in normal numbers, but exhibited clumping of their dendritic fields, indicative of a failure of self-avoidance [44, 45].

The parallel molecular and phenotypic diversity of cPcdhs could, *a priori*, indicate three potential models, all of which may be correct for distinct subsets of neurons or distinct functions. First, isoform diversity *per se* may be required. Second, there may be a high level of isoform redundancy such that any one (or few) isoform(s) may suffice. Third, there may be unique roles for *individual* isoforms, such that no other isoform can compensate for their loss. Some evidence exists for each of these possibilities. The γC3-5 isoforms, but not γA1-3, were required for postnatal viability in mice [34]. As γC5 is expressed later in the postnatal period [46], and γC4 is unable to localize to the cell membrane alone (it requires carrier isoforms from among the β- and other γ-Pcdhs) [17, 18, 47], it seemed most likely that this reflected a crucial role for γC3. While re-expressing a single γ-Pcdh isoform (γA1 or γC3) could rescue self-avoidance in SACs, it disrupted self/non-self recognition required for proper receptive field overlap and retinal circuit performance [44, 45]. Similar re-expression of γA1 or γC3 in cortical neurons on a *Pcdhg* null background led to aberrantly increased or decreased dendrite arborization, depending on whether surrounding cells also expressed the same single isoform and could thus presumably engage homophilically [40]. A requirement for cPcdh diversity *per se* was discovered for the convergence of olfactory sensory neuron axons on particular glomeruli in the olfactory bulb: disruption of olfactory circuitry was mild in single gene cluster mutants, but devastating in mice lacking all three clusters, and it was not rescued by re-expressing a triad of single α-, β-, and γ-Pcdh isoforms [5, 48]. Support for unique roles for individual cPcdh isoforms comes from the demonstration that the disrupted axonal branching observed in *Pcdha* mutant serotonergic neurons [8] is due entirely to the role of one isoform, αC2 (the dominant isoform expressed in these neurons) [4], and from the demonstration of a unique role for γC3 in regulating Wnt signaling through Axin1 [49].

Thus, no single model is likely to encompass all of the γ-Pcdhs' diverse functions. Additionally, most prior studies have relied on mis- or over-expression of individual isoforms, which could in some cases result in new, distinct phenotypes. In the first part of the present study, and paralleling approaches establishing the necessity of *Dscam1* molecular diversity for specific neurodevelopmental roles in *Drosophila* [23–26], we used CRISPR/Cas9 genome editing to simultaneously target the 22 *Pcdhg* variable exons in an unbiased manner, creating a new allelic series of mouse mutants with reduced isoform diversity from the endogenous gene cluster. In the second part of the present study, we find that only one isoform, γC4, is strictly required for postnatal viability and survival of the many neuronal subsets shown previously to depend on the γ-Pcdhs. Our results: 1) show that some γ-Pcdh functions do not require molecular diversity; 2) confirm that at least some γ-Pcdh isoforms have unique roles; and 3) suggest that the regulation of neuronal survival may require novel mechanisms of cPcdh interaction and/or signaling involving γC4.

## Results

### CRISPR/Cas9 strategy for reducing *Pcdhg* isoform diversity

To assess the importance of γ-Pcdh isoform diversity to postnatal viability and neurodevelopmental functions, we used a shotgun CRISPR/Cas9 genome editing screen to disrupt varying numbers of *Pcdhg* variable exons (Fig 1B–1D). We reasoned that by injecting pooled individual single guide RNAs (sgRNAs) targeting each V exon into many zygotes, we could generate a number of unique mutant mouse lines, each of which harbored distinct patterns of reduced isoform diversity due to variability in which sgRNAs efficiently bound and directed mutations. We designed sgRNAs to target within ~100–150 base pairs downstream of the start codon of each *Pcdhg* V exon, with the goal of creating frame-shifting mutations through non-homologous end joining (NHEJ) repair (Fig 1B and 1C)[50, 51]. A total of twenty guides were designed; the *PcdhgB4* and *PcdhgB5* V exons shared enough homology to allow a single guide to target both, as did *PcdhB6* and *PcdhgB7*. Pooled sgRNAs were concentrated, then combined with Cas9 mRNA and microinjected into C57BL/6J mouse zygotes at three different concentrations (each individual guide at 50 ng sgRNA/µl, 10 ng sgRNA/µl, or 5 ng sgRNA/µl: HI, MED, and LO, respectively), as described in Materials and Methods (Fig 1D). Microinjected zygotes were transferred into a total of 20 pseudopregnant females, resulting in 100 live-born mice (16 from HI, 34 from MED, and 50 from LO). All 100 founders were screened at 7 *Pcdhg* exons by PCR with Sanger sequencing to detect frame-shifting mutations. From this initial screen, 15 founders exhibited some disruption (10 from HI and 5 from MED) and were designated for breeding. Most disruptions were found in pups from HI or MED injections, so the 50 founders resulting from the LO injections were not pursued further. The remaining 35 founders were further screened by PCR and Sanger sequencing at the remaining 15 *Pcdhg* variable exons. In this way, a total of 31 founders were identified that carried some constellation of mutations at the guide-targeted sites (14/16 HI, 17/34 MED).

Due to the likely mosaicism of the founders and the uncertainty of germline transmission of any given mutation, we did not characterize the 31 identified founders more extensively. Rather, each was crossed with wild-type C57BL/6J animals to generate G1 offspring for further analysis (Fig 1D). Sperm from male G1 mice was cryopreserved while somatic tissue was used for genotyping to identify lines of interest carrying reduced diversity of *Pcdhg* variable exons. Ninety-four G1 offspring were screened for heterozygous mutations using a custom amplicon assay from Illumina and Illumina MiSeq sequencing (see Materials and Methods for details, target coordinates are listed in the S1 File). The amplicon assay was designed to sequence the 22 *Pcdhg* sgRNA target regions, all the analogous regions in the untargeted *Pcdha* and *Pcdhb* clusters, and the top 95 predicted potential off-target sites (Fig 1D). We screened for missense and nonsense mutations, small indels and, because our approach was intended to produce up to 22 double-stranded breaks within 160 kilobases, larger rearrangements between breakpoints. Smaller indels were identified using the Genome Analysis Toolkit (GATK) [52], while larger rearrangements were identified with BreaKmer [53]. Twenty-six distinct lines carrying unique constellations of mutations are represented in Table 1, derived from 12 different founders (nucleotide sequences for mutations are listed in the S2 File). Most mouse lines (20 lines from 9 founders) resulted from microinjection of the highest concentration of total sgRNAs (50 ng/µl/guide). Mutations ranged from only one disrupted V exon (leaving 21 intact) to 21 disrupted V exons (leaving only 1 intact; Table 1). Each V exon was disrupted in at least one mouse line, either by discrete indels, or through involvement in a rearrangement between breakpoints.

**Table 1. Summary of *Pcdhg* exons disrupted in CRISPR/Cas9 targeted strains.** Individual mouse lines are summarized with the allele name along with the name used to describe the lines analyzed here. Red boxes indicate disrupted exons–"X" for discrete frame-shifting indel and dashed lines for large scale rearrangements or deletions between the indicated exons. "IF" indicates an in-frame indel predicted to result in an expressed protein.

| Allele | Name here | A1 | A2 | A3 | B1 | A4 | B2 | A5 | A6 | A7 | B4 | A8 | B5 | A9 | B6 | A10 | B7 | A11 | B8 | A12 | C3 | C4 | C5 | Intact exons |
|---|---|---|---|---|---|---|---|---|---|---|---|---|---|---|---|---|---|---|---|---|---|---|---|---|
| *Pcdhg^em4* | | | | X | X | | | X | | | | | | | | | | | | | | | | 19 |
| *Pcdhg^em5* | 13R1 | X | | | X | | | X | | X | | | | [− | — | −] | X | | | | | X | | 13 |
| *Pcdhg^em6* | | [− | — | −] | | IF | | | | | | | | | | X | IF | X | IF | | | | | 17 |
| *Pcdhg^em7* | | | | X | | | | | | | | | | | | X | | | | | | X | | 19 |
| *Pcdhg^em8* | 1R1 | [− | — | — | — | — | — | — | — | — | — | — | — | — | — | — | — | — | — | −] | | X | | 1 |
| *Pcdhg^em9* | | [− | — | — | — | — | — | −] | | IF | | | X | | IF | | | | | X | | | | 13 |
| *Pcdhg^em10* | | | | | | | | IF | | | | | | | | X | | | | | IF | IF | X | 20 |
| *Pcdhg^em11* | | IF | | | | | | X | | | | | | | | | | | | | | | | 21 |
| *Pcdhg^em12* | 3R1 | [− | — | — | — | −] | X | [− | — | — | — | — | — | — | — | — | — | — | — | −] | | | | 3 |
| *Pcdhg^em13* | | X | X | X | | | | | | | | | X | | | | | | | | | | | 18 |
| *Pcdhg^em14* | | X | | X | | | | X | | | | X | | | | | | | | | | | | 18 |
| *Pcdhg^em16* | | | | | | | | | | | | | X | X | | | | | | | | X | | 19 |
| *Pcdhg^em17* | | [− | — | — | — | — | — | −] | | | | | | | IF | | | | | X | X | | | 13 |
| *Pcdhg^em19* | | | | | [− | — | — | — | — | — | — | — | — | — | — | — | — | — | −] | X | | | | 6 |
| *Pcdhg^em23* | | [− | — | — | — | — | — | — | — | −] | | | | | | | | | | | | | | 13 |
| *Pcdhg^em24* | | | | | | | | | | | | | | | | | | X | | | | | | 21 |
| *Pcdhg^em25* | | | X | X | | | | | | | | | | | | X | [− | — | −] | | | | | 16 |
| *Pcdhg^em27* | | [− | — | — | — | — | — | — | — | — | −] | | | | | | | | | | | | | 12 |
| *Pcdhg^em28* | | X | | X | | | | | | | | | | | | | X | | | | | | | 19 |
| *Pcdhg^em31* | | | | | | | X | | | | | | | | | | | X | | | IF | IF | X | 19 |
| *Pcdhg^em32* | | | | | IF | [− | — | — | — | −] | X | | | | | | | | | | | | | 16 |
| *Pcdhg^em33* | | X | X | [− | — | — | — | −] | | | [− | — | — | — | — | — | — | — | −] | X | | | | 5 |
| *Pcdhg^em34* | | [− | — | — | — | — | — | −] | | | | | IF | | IF | X | | | | | | | | 14 |
| *Pcdhg^em35* | 3R2 | [− | — | — | — | — | — | — | — | — | — | — | — | — | — | — | — | — | −] | X | | IF | X | 3 |
| *Pcdhg^em42* | | X | | | | | | | | | | | IF | | | | | | | | | | | 21 |
| *Pcdhg^em44* | | | | | | | | | | | | | | | | | | | | | X | | | 21 |
| *Pcdhg^em41* | C4KO | | | | | | | | | | | | | | | | | | | | | X | | 21 |

## Whole genome sequencing reveals rearrangements undetected by amplicon sequencing

Based on initial results from the custom amplicon assay, we chose three lines for cryo-recovery and further analysis–*Pcdhg^em5*, *Pcdhg^em12*, and *Pcdhg^em35* (Table 1). Mice from each recovered line were intercrossed to generate homozygous mutants. Once obtained, we used these homozygous mutants to verify by PCR V-exon indels and rearrangements. We found that fewer exons were successfully amplified by PCR than expected (S1A Fig), indicating that amplicon sequencing failed to detect all the mutations harbored by the heterozygous G1 mutants. Therefore, we performed whole genome linked-read sequencing on homozygous mutants, using the Chromium Genome Sequencing Solution from 10X Genomics to detect large scale rearrangements (S1 Fig, S2 Fig). Paired-end reads from the whole genome sequencing were used to reconstruct the rearrangements (Fig 2). We found that *Pcdhg^em5* contained frame-shifting indels in V exons A1, B1, A5, A7, B7, and C4, all of which were accurately called by the prior amplicon sequencing. However, there was an inversion and deletion that disrupted exons A9, B6, and A10 that was undetected by the previous analysis (S2A Fig). Thus, *Pcdhg^em5* retained 13 intact V exons (Fig 2A), and we renamed the allele *Pcdhg^13R1*; "13R" for the number of intact exons remaining and "1" as it was the first allele identified with this number (referred to as 13R1 hereafter for simplicity).

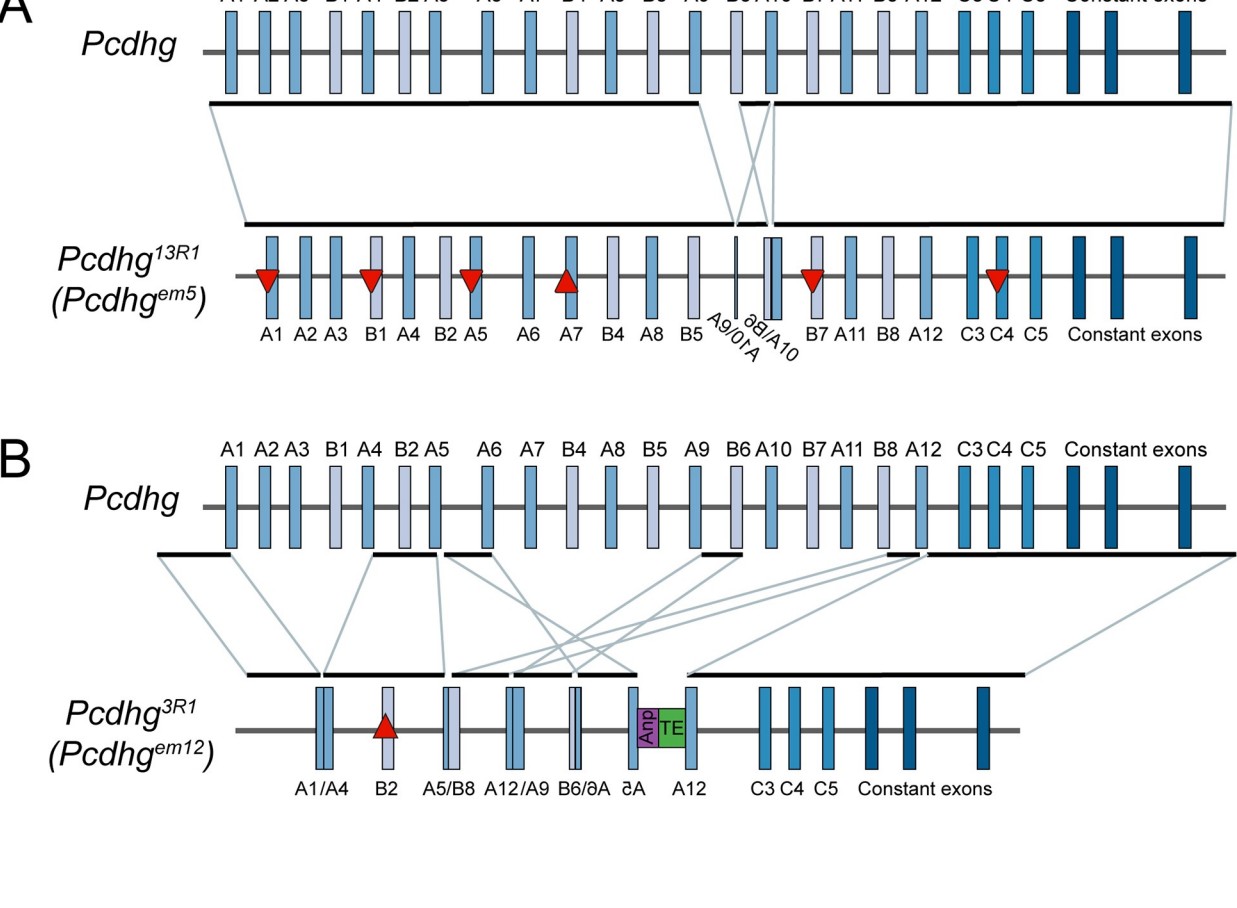

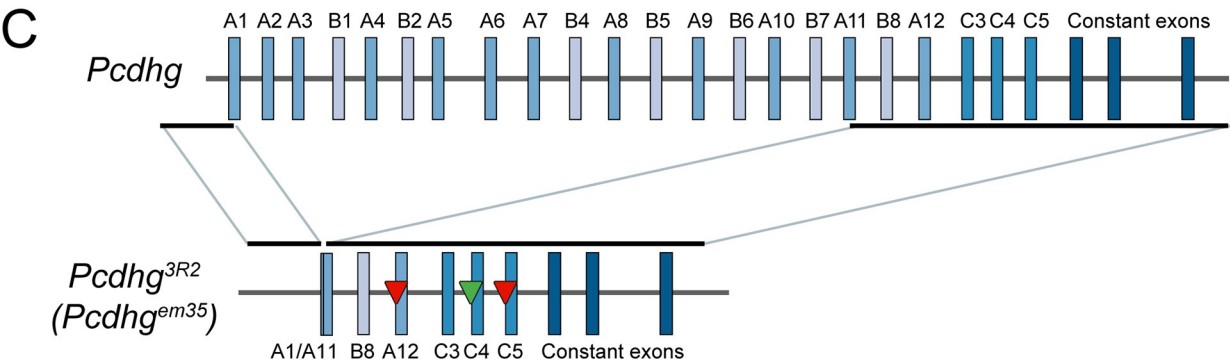

**Fig 2. Three new *Pcdhg* alleles with reduced isoform diversity *via* CRISPR/Cas9 genome editing.** Schematics of the mutated *Pcdhg* alleles verified or identified by whole genome sequencing in three strains illustrate the range of mutations induced by CRISPR/Cas9 targeting. **A**) In 13R1 mutants, frame-shifting indels disrupted 6 exons (upward triangle indicates insertion, downward indicates deletion, red indicates frame-shift) and an inversion and deletion disrupted exons A9, B6, and A10. **B**) In 3R1 mutants, there was a small frame-shifting insertion in exon B2 along with multiple deletions and rearrangements between exons. For example, genomic DNA between the exon A5 and exon A6 guide sites was inverted and inserted at the B6 guide site, followed by coding sequence from the gene Anp32a (purple box) and a transposable element (green box). Only exons C3, C4, and C5 remain intact. **C**) In 3R2 mutants, a single large deletion resulted in a fusion between guide sites at exon A1 and exon A11. This was accompanied by small, frame-shifting deletions in exons A12 and C5 and a small in-frame deletion in exon C4 (downward pointing green triangle).

Sequencing of *Pcdhg^em12^* revealed extensive rearrangements. A fusion between exons A5 and B8 identified by the amplicon sequencing was confirmed, but additional junctions were found between A1 and A4, between A12 and A9, and between B6 and A6 (Fig 2A, S1C Fig).

Furthermore, linked reads revealed an insertion including a transposable element and coding sequence from *Anp32a* which aligns to exons 4–7 of transcript Anp32a-201 without the intervening introns (Fig 2B, S1D Fig). This phenomenon of the insertion of a transposable element along with coding sequence from an early expressed gene has been previously described in CRISPR genome editing [54]. As this sequence was inserted 3' to the inverted exon *Pcdhga5*, there is no associated transcription start site, and no protein product is expected. Altogether, at least 9 double-stranded breaks occurred, resulting in a frame-shifting 52 bp insertion into exon B2, and larger deletions and rearrangements disrupting all the other γA and γB isoforms. As only the 3 γC V exons remained intact in *Pcdhg^em12^*, we renamed this allele *Pcdhg^3R1^* (3R1 hereafter; Fig 2B).

Whole genome sequencing from *Pcdhg^em35^* mutants confirmed small frame-shifting deletions at exons A12 and C5, as well as an in-frame deletion of 9 bp in exon C4. However, this more exhaustive sequencing also revealed a large rearrangement undetected by the prior amplicon sequencing analysis. Here, there was a ~94 kb deletion spanning the breakpoint from exon A1 to that of exon A11 (S2B Fig). Only exons B8 and C3 were unaffected by any mutation. As exon C4 still encoded a nearly full-length protein lacking only 3 amino acids (residues 27–29 in the signal peptide), we renamed the *Pcdhg^em35^* allele *Pcdhg^3R2^* (3R2 hereafter, Fig 2C), as it represented the second allele identified with 3 isoforms remaining.

In all three mutants analyzed thus far, rearrangements were identified by whole genome sequencing that were undetected by the amplicon analysis (Fig 2, S1 Table). With this information, we re-analyzed the amplicon sequencing data by visual inspection of the paired reads within the Integrative Genomics Viewer (IGV). We were able to find many, but not all, of the junctions identified by the whole genome sequencing, but missed by BreaKmer analysis of the amplicon sequencing (S1B Fig). Therefore, we manually inspected the alignments from each of the other frozen lines. Additional rearrangements were identified where each of the paired ends of multiple reads mapped to different exons (S1 Table).

## Isoform diversity *per se* is not required for postnatal viability

The complete deletion of the Pcdhg cluster results in neonatal lethality [33], as does the deletion of the three γC V exons, whereas mice with deletion of the first three γA exons (A1, A2, A3) had no reported phenotypes [34]. Of the three lines initially characterized, 3R1 and 3R2 homozygous mutant mice were born at expected Mendelian ratios and survived into adulthood without any overt differences from their wild-type or heterozygous littermates. In contrast, 13R1 homozygous mutants died within hours of birth with the hunched posture, tremor, and inability to right themselves or nurse that is characteristic of *Pcdhg^del/del^* mutants lacking the entire *Pcdhg* cluster [33, 37]. Whole genome sequencing from all three lines confirmed that there were no off-target mutations in the *Pcdha* or *Pcdhb* locus, or any disruptions in the *Pcdhg* constant exons. However, it remained formally possible that the particular rearrangements within 13R1 mutants disrupted the expression of the other isoforms or V exon splicing to the constant exons, creating a functional null mutation.

To exclude this possibility, we performed quantitative real-time PCR using cDNA from the cerebral cortices of homozygous mutants and wild-type littermates (Fig 3A). To detect specific isoform transcripts, forward primers targeting the 3' end of each V exon were used with a reverse primer in constant exon 1 (product spans 1 intron) or exon 2 (product spans 2 introns). To monitor total levels of *Pcdhg* locus transcription, a forward primer in constant exon1 was used with a reverse primer in constant exon 2 (spanning 1 intron, primer sequences in S2 Table). Total locus expression was significantly reduced in 3R2 homozygous mutants, but in both 3R1 and 13R1 mutant cortex, expression differences did not reach statistical significance. Furthermore, individual isoform transcription was reduced significantly only when

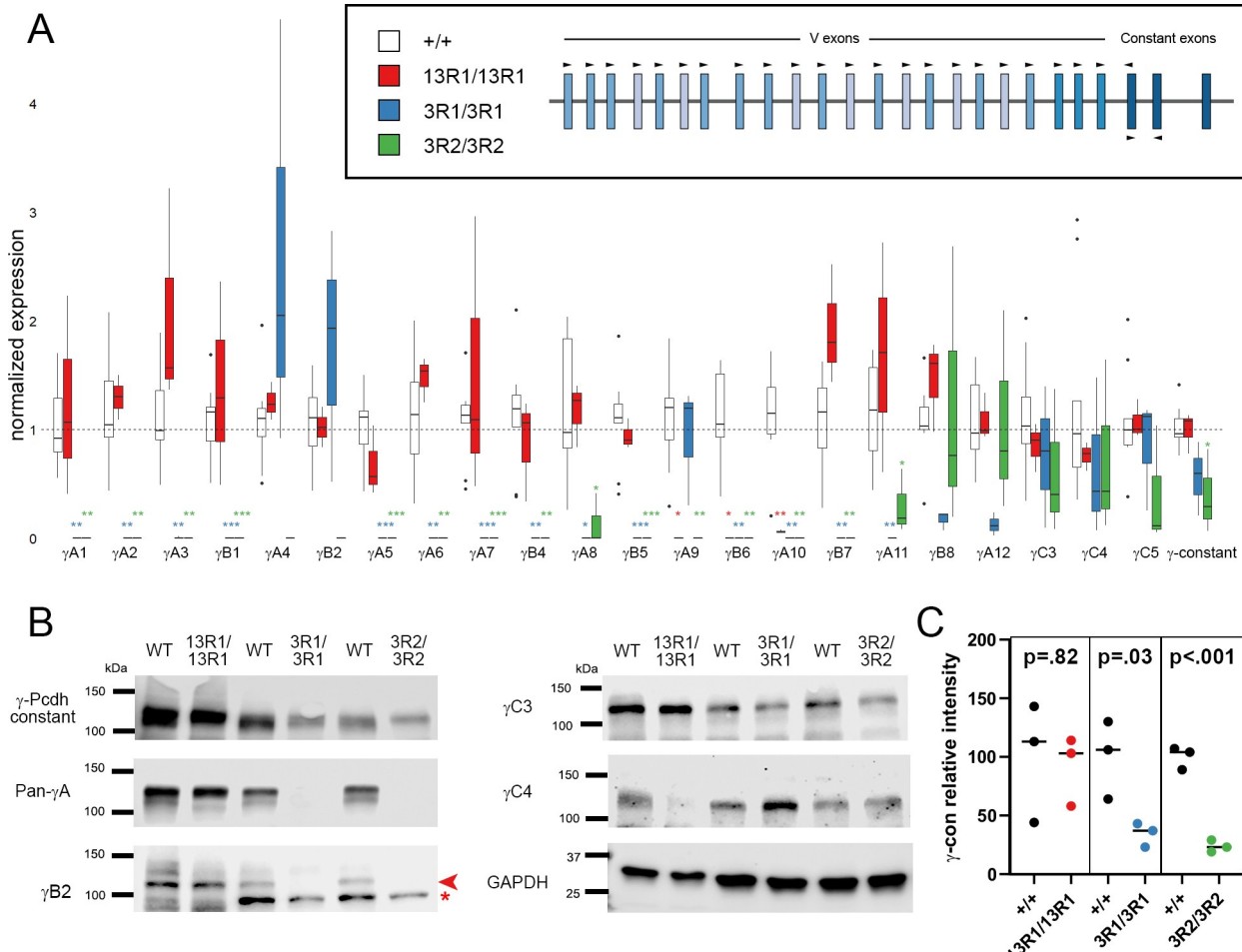

**Fig 3. *Pcdhg* isoform expression reflects genomic locus status in CRISPR/Cas9 mutant lines. A**) Quantitative RT-PCR analysis of cerebral cortex cDNA from 13R1 mutants (red), 3R1 mutants (blue) and 3R2 mutants (green) verified that intact isoforms were expressed at expected levels. Expression of constant exons was significantly reduced in 3R2 mutants only. 13R1 mutants were analyzed at P0, while 3R1 and 3R2 animals were analyzed at P14, each with littermate controls (white). * = p < 0.05; ** = p < 0.01; *** = p < 0.001 by Tukey post-hoc test comparing the indicated genotype with wild type. n = 3–9 animals per genotype. Box plots represent the median, first and third quartile, range, and outliers. **B**) Western blots of brain lysates from 13R1 (at P0), 3R1, and 3R2 mutants (at 1 month of age) with littermate controls confirm the protein isoform expression predicted by allele sequencing. Arrowhead indicates the specific γB2 band at the predicted size; asterisk indicates non-specific background band of incorrect size present in adult samples only. **C**) Total γ-Pcdh protein levels were quantified using Li-Cor quantitative western blotting, normalizing to GAPDH. Values from three samples per genotype were compared to littermate controls using student's t-test with the indicated p-values.

mutations completely disrupted the exon by deletion or inversion. Smaller indels generally had no effect on transcript expression level (e.g., A1, B1, A5, A7, B7, and C4 were not significantly reduced in 13R1 homozygous mutants, but A9, B6, and A10 were undetectable). Additionally, transcription of several V exon fusions was detected, including A4 (fusion with A1) and A9 (fusion with A12) in 3R1 (predicted protein products encoded by these fused transcripts are listed in S2 File). We also asked the extent to which mutations within the *Pcdhg* locus altered isoform transcription from the *Pcdha* or *Pcdhb* clusters. While there were no significant changes detected in 13R1 mutants, isoforms from the 3' end of the *Pcdhb* locus were expressed at significantly higher levels in 3R1 and 3R2 homozygous mutants compared to controls (β11 in 3R1, β15 and β22 in both 3R1 and 3R2). Expression of *Pcdha* cluster genes appeared to be unchanged (S3 Fig).

We also verified that these mRNA expression levels were reflected at the protein level, utilizing a series of antibodies specific for particular γ-Pcdh protein isoforms [55]. Western blot analysis of brain lysates from 13R1 neonates and 3R1 and 3R2 adults confirmed the presence of the expected isoforms at the appropriate molecular weights: 13R1 brains expressed γA isoforms, γB2, and γC3, but not γC4, while 3R1 and 3R2 brains expressed γC3 and γC4, but not γB2 or any γA isoforms (Fig 3B). We used quantitative western blotting to compare the levels of total γ-Pcdh constant domain with littermate controls (Fig 3C). 13R1 mutants were indistinguishable from wild type littermates, while 3R1 and 3R2 mutants were significantly reduced (despite the fact that differences did not reach statistical significance at the RNA level, Fig 3A). Based on these analyses, we concluded that 13R1 homozygous mutants are not, in fact, complete *Pcdhg* functional nulls, and that the similarity of their neonatally lethal phenotype to that of nulls likely reflects the essential nature of a particular isoform lost in this line but present in both 3R1 and 3R2; that is, γC4, as described below. Before testing this conclusion in detail, we first asked whether the neonatal lethality of 13R1 was accompanied by cellular phenotypes previously described in *Pcdhg*<sup>del/del</sup> null mice, and whether the viable 3R1 and 3R2 lines lacked these phenotypes.

## Excessive developmental neuronal apoptosis occurs in mutants exhibiting neonatal lethality

*Pcdhg*<sup>del/del</sup> mutants exhibit neonatal lethality with excessive cell death of interneurons in the ventral spinal cord and brainstem, accompanied by reactive gliosis [31, 33, 37]. To ask if lethality in reduced diversity mutants was also accompanied by increased developmental apoptosis, we analyzed cryosections from spinal cords at P0. Indeed, we found significant cell death in 13R1 homozygous mutants, but not 3R1 or 3R2 mutants (Fig 4). Staining of transverse sections for the pan-neuronal marker NeuN revealed that 13R1 mutant spinal cords were grossly smaller, with obvious reductions in cell number primarily in the ventral spinal cord, as reported previously for *Pcdhg*<sup>del/del</sup> null mutant neonates (Fig 4A–4C) [31, 33]. This was accompanied by an increase in reactive astrocytes, revealed by GFAP labeling within the gray matter (Fig 4D–4F), which we showed previously in null mice to be a response to neuronal apoptosis [37]. Spinal interneurons derive from 6 dorsal (dI1-6) and 4 ventral (V0-3) domains (reviewed by [56]); we used antibodies against two transcription factors, FoxP2 and Pax2, that label distinct subsets of interneurons. We found significantly fewer FoxP2-positive and Pax2-positive ventral interneurons in 13R1 mutants than in wild type littermates or 3R1 homozygous mutants (Fig 4G–4Q). To confirm that cell loss resulted from excessive apoptosis as observed in *Pcdhg* nulls, we assayed for cleaved caspase 3 (CC3), a marker of apoptotic cell death, and found significantly more CC3-labeled profiles in 13R1 mutants than in wild type littermates or 3R1 (Fig 4M–4O and 4R). An additional phenotype previously described in *Pcdhg* null mutants is the clumping of parvalbumin-positive Ia afferent axon terminals around their motor neuron targets in the ventral horn; this phenotype is worsened by, though not entirely due to, increased interneuron apoptosis [36]. Again, we found that 13R1 mutants exhibited a null mutant-like phenotype while 3R1 mutant Ia afferent projections appeared similar to those of controls. (S4 Fig).

Increased cell death of many neuronal subtypes is also a hallmark of *Pcdhg* loss of function in the retina, with retinal cell death occurring largely postnatally [10, 43]. To circumvent the neonatal lethality of 13R1 mutants, we made compound heterozygous mutants with the conditional loss of function allele *Pcdhg*<sup>fcon3</sup> [31] crossed with Pax6α-Cre to restrict recombination of this allele to the retina [57]. These mutants are referred to as 13R1/cRKO (for conditional retinal knockout). Examination of immunostained retinal cross sections from 13R1/cRKO

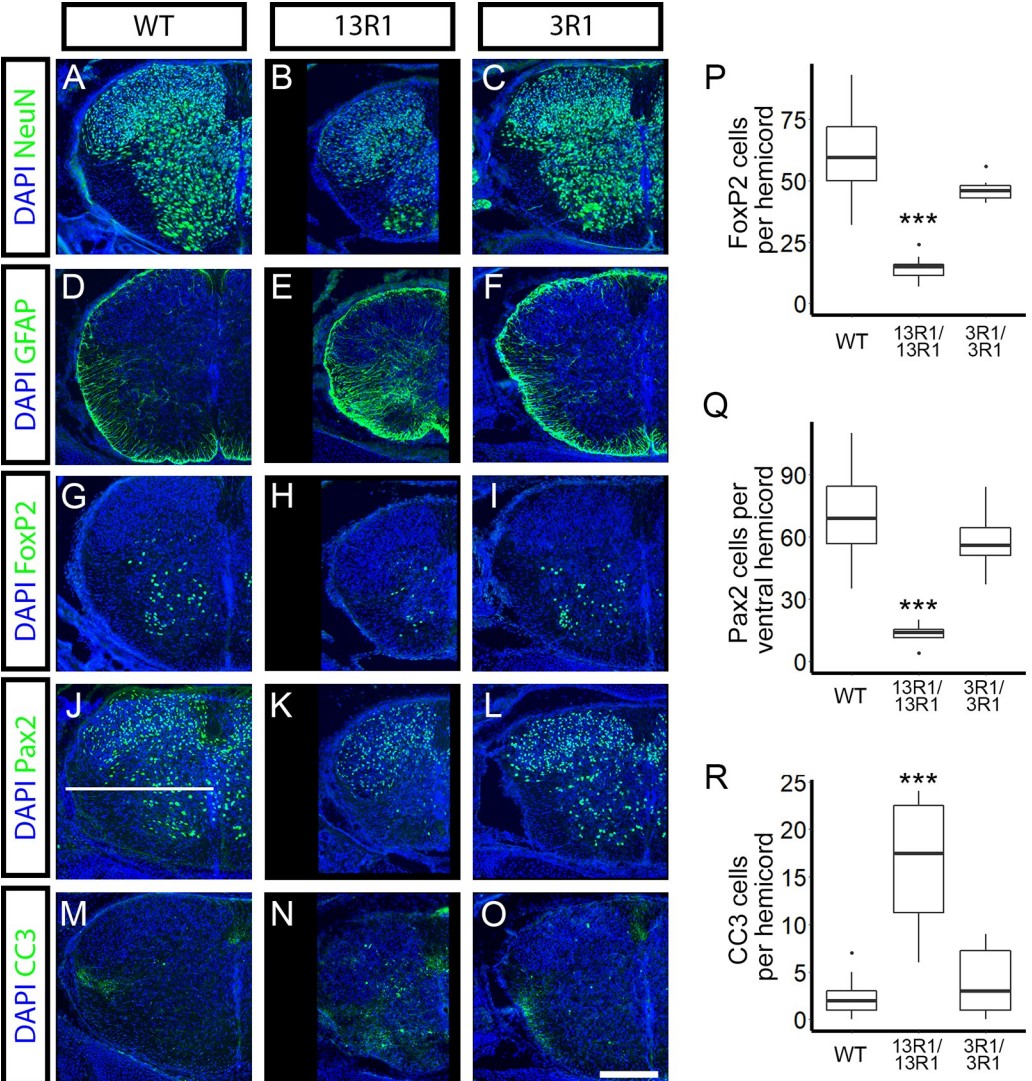

**Fig 4. Spinal interneurons undergo excessive apoptosis in 13R1, but not 3R1 mutants.** Cryosections from P0 spinal cords were immunostained for the indicated markers (green; DAPI counterstain for nuclei, blue). NeuN staining in cryosections from P0 spinal cords of (**A**) wild type, (**B**) 13R1 mutants, and (**C**) 3R1 mutants revealed that 13R1 mutant cords were smaller with substantial loss of ventral interneurons. (**D-F**) Comparison of GFAP-positive astrocytes within the ventral spinal cord showed reactive gliosis in (**E**) 13R1, but not in (**D**) wild type or (**F**) 3R1. Specific interneuron populations were quantified, including (**G-I,P**) FoxP2+ and (**J-L,Q**) Pax2+ ventral interneurons. Both populations were significantly reduced in 13R1 mutants compared to wild type or 3R1 animals (**P**, **Q**). The horizontal white line in **J** indicates the divide used to quantify ventral interneurons. **M-O**) Staining for cleaved caspase 3 (CC3) reveals more apoptotic cells in (**N**) 13R1 than in (**M**) wild type or (**O**) 3R1 mutants (quantified in **R**). Scale bar is 200 μm. ** = p < 0.01; *** = p < 0.001 by Dunnett's multiple comparison test comparing the indicated genotype with wild type. n = 32–48 hemicords total from 3 animals per genotype. Box plots represent the median, first and third quartile, range, and outliers.

mutants at P14 revealed substantial thinning of the inner retina, including both cellular and synaptic layers, compared to wild type (Fig 5A and 5B), whereas 3R2 mutant retinas were not thinner (Fig 5C). As reported for *Pcdhg* null mutants [43], the retinal thinning in 13R1/cRKO mice was not accompanied by obvious disorganization of neurite stratification within the inner plexiform layer (Fig 5D–5I). To verify that this resulted from cell loss, we measured the density of two amacrine cell types (tyrosine hydroxylase (TH)+ dopaminergic amacrine cells and VGLUT3+ amacrine cells) and two retinal ganglion cell types (Melanopsin+ RGCs and

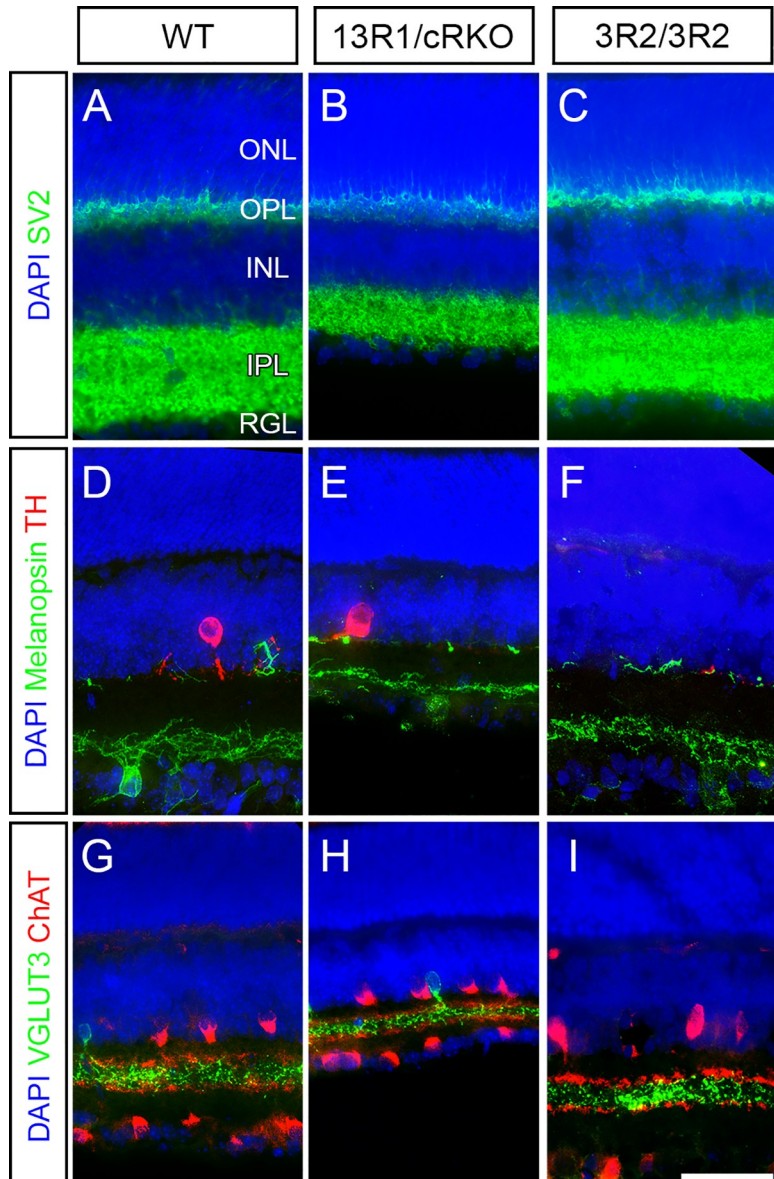

**Fig 5. Substantial loss of retinal thickness without layer disorganization in 13R1 mutants. A-C**) Cryosections taken in cross section through retinas of (**A**) wild type, (**B**) 13R1/cRKO, and (**C**) 3R2 mutants at two weeks of age were immunostained for SV2 to label the synaptic layers (outer and inner plexiform layers; OPL, IPL) and DAPI to label the inner and outer nuclear layers (ONL, INL) and retinal ganglion cell layer (RGL). The IPL and INL were both notably thinner in 13R1/cRKO than in wild type or 3R2 mutants. This was not accompanied by disorganization of neuronal subtype stratification within the IPL, as revealed by (**D-F**) Melanopsin and TH immunolabeling of ipRGCs and dopaminergic amacrine cells, respectively, and (**G-I**) VGLUT3 and ChAT immunolabeling of glutamatergic amacrine cells and starburst amacrine cells, respectively. Scale bar is 50 μm. Images are representative of at least 6 retinas per genotype analyzed.

Brn3a+ RGCs) in whole-mount retinas from P14 animals. All four cell types were significantly less numerous in 13R1/cRKO mutants than in wild-type littermates or in 3R1 or 3R2 homozygous mutants, neither of which exhibited any abnormalities (Fig 6). Together our results show that the excessive neuronal apoptosis and postnatal lethality observed in *Pcdhg* mutants does not depend on isoform diversity *per se*.

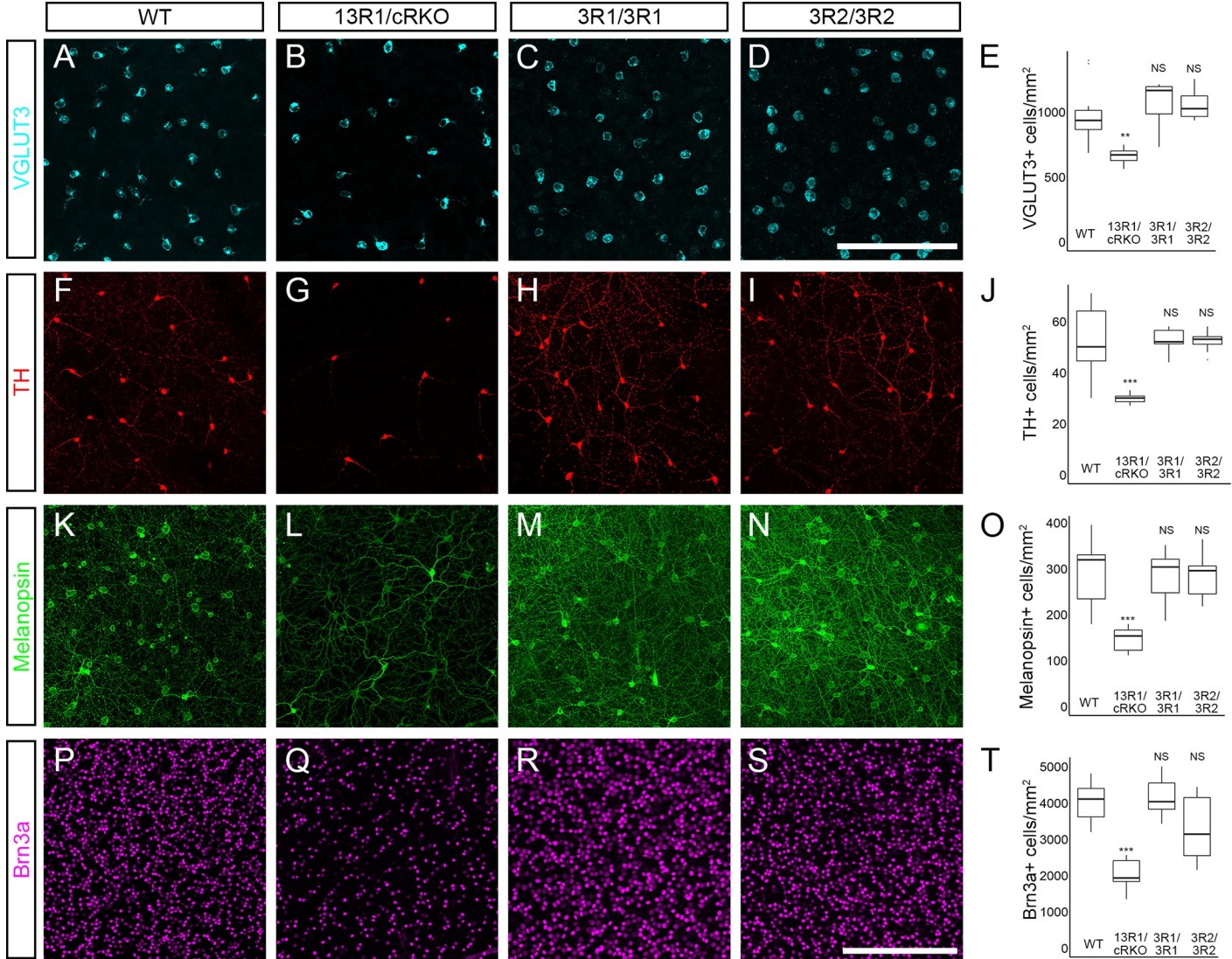

**Fig 6. Reduced retinal neuron numbers in 13R1 mutants.** Whole mount retinas from two-week-old mice were immunostained to label neuronal subtypes, imaged *en face* by confocal microscopy, and cell densities quantified. Analyzed cell types include (**A-E**) VGLUT3+ amacrine cells, (**F-J**) TH+ dopaminergic amacrine cells, (**K-O**) Melanopsin+ ipRGCs, and (**P-T**) Brn3a+ RGCs. For all four of the subtypes assayed, densities were significantly reduced in 13R1/cRKO retinas compared to wild-type. Densities were not reduced in either 3R1 or 3R2. Scale bar in **D** is 100 μm (for **A-D**); scale bar in **S** is 300 μm (for **F-S**). ** = $p < 0.01$; *** = $p < 0.001$ by Tukey post-hoc test comparing the indicated genotype with wild type. n = 6 retinas per genotype. Box plots represent the median, first and third quartile, range, and outliers.

## *Pcdhgc4* is the necessary and sufficient *Pcdhg* cluster gene for postnatal viability

Having confirmed that the phenotypes of 13R1 resemble the complete deletion of the *Pcdhg* cluster, in the second part of the study we turned to utilizing the novel CRISPR mutant mouse lines to confirm which γ-Pcdh isoforms were critical. We noted that homozygous 3R1 mutants survived and did not exhibit exacerbated apoptosis despite lacking expression of any functional γA or γB isoforms, indicating that one or more of the C3-C5 isoforms must be critical. Furthermore, 3R2 mutants survived and were phenotypically normal without a functional *Pcdhgc5* gene, whilst 13R1 homozygous mutants, which exhibited neonatal lethality and exacerbated neuronal apoptosis uniquely harbored frame-shifting mutations in *Pcdhgc4* with no

changes in expression level or coding sequence of either *Pcdhgc3* or *Pcdhgc5* (Fig 3, Table 1). In a separate study, we have derived and are analyzing a CRISPR-targeted mouse line that specifically generated a *Pcdhgc3* loss-of-function allele, and have found that they are viable and fertile as adults (S5 Fig, D. Steffen, K.M. Mah, P.J. Bosch, A.M. Garrett, R.W. Burgess, and J.A. Weiner, in preparation). Together with prior data indicating that mice lacking the entire *Pcdha* and *Pcdhb* clusters are viable [5, 48], this suggests that *Pcdhgc4* encodes the sole cPcdh isoform essential for organismal survival. We generated two additional mouse lines to confirm this conclusion.

First, we chose *Pcdhg^em8* from our list of mutants for cryorecovery (Table 1). This line harbored a large deletion from within exon A1 to within exon C3 identified by visual inspection of the aligned reads from the original amplicon sequencing, as well as a 1 bp frame-shifting deletion in exon C5. Upon cryorecovery, we generated homozygous mutants and verified these mutations by linked-read whole genome sequencing (Fig 7A, S6A Fig). As only V exon C4 was left intact, this strain was renamed *Pcdhg^1R1* (1R1 hereafter). These homozygous mutants survived into adulthood and were fertile, with no overt differences from their wild type litter mates. Both quantitative RT-PCR and western blot analyses reflected the expected isoform expression, while levels of total γ-Pcdh were reduced at the mRNA and protein level (Fig 7C, S7 Fig).

Second, we generated an entirely new mutant mouse line by specifically targeting *Pcdhgc4* for CRISPR/Cas9-directed gene disruption. This resulted in a 13 bp frame-shifting deletion 3' to the start codon (Fig 7B; S6B Fig). We named this new allele *Pcdhg^C4KO* (C4KO hereafter). Homozygous C4KO mutants exhibited the same hunched posture, tremor, and inability to nurse or right themselves observed in *Pcdhg^del/del* and 13R1 mutants and died shortly after birth. Sanger sequencing confirmed that the analogous regions of the other *Pcdhg* V exons were not disrupted in this line (S6B Fig), while western blot analyses verified that the other isoforms were produced as expected (Fig 7C). Together, these two additional mouse lines show that mutation of *Pcdhgc4* alone recapitulated the overt phenotype of losing the entire cluster, whilst expression of this single γ-Pcdh isoform was sufficient to rescue viability, even when all 21 other *Pcdhg* isoforms were absent.

## *Pcdhgc4* is the crucial isoform for neuronal survival

To ask if these overt phenotypes of death *vs.* survival in C4KO and 1R1 mutants extended to neuronal cell survival, we again analyzed spinal cords from P0 neonatal mutants. As expected from their outward appearance, 1R1 homozygous spinal cords appeared grossly normal in overall size and neuronal density (NeuN+ cells), with little if any reactive gliosis (GFAP+; Fig 8B and 8D). In contrast, C4KO mutant spinal cords were grossly smaller and exhibited clear ventral interneuron loss (Fig 8A) accompanied by reactive gliosis (Fig 8C) that was essentially identical to that observed in complete *Pcdhg* null mutants [31, 33] and 13R1 mutants (Fig 4). Analysis of individual cell types revealed significant loss of FoxP2-labeled cells and Pax2-positive neurons in the ventral spinal cord of C4KO mutants (Fig 8E, 8G, 8K and 8L). Significant reductions in 1R1 mutant cell density were observed, though they were much more modest (Fig 8F, 8H, 8K and 8L). Consistent with this, there were significantly more CC3-positive neurons in C4KO mutants than in 1R1 or WT neonates (Fig 8I, 8J and 8M). As expected, these patterns of spinal interneuron survival corresponded to the presence (C4KO; S8A Fig) or absence (1R1; S8B Fig) of aggregated Ia afferent terminals around motor neurons in the ventral horn.

As 1R1 mutants survive into adulthood, we analyzed neuronal survival in the retina at 14 days of age and in adult. There was no indication of retinal thinning at either age. As above,

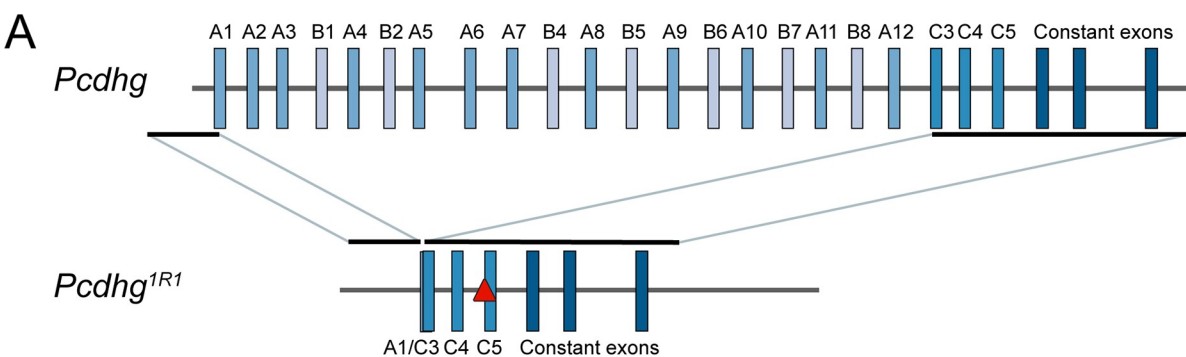

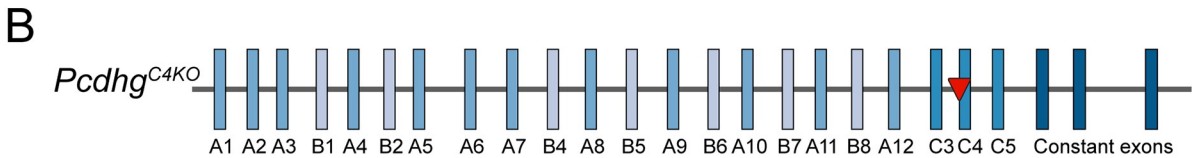

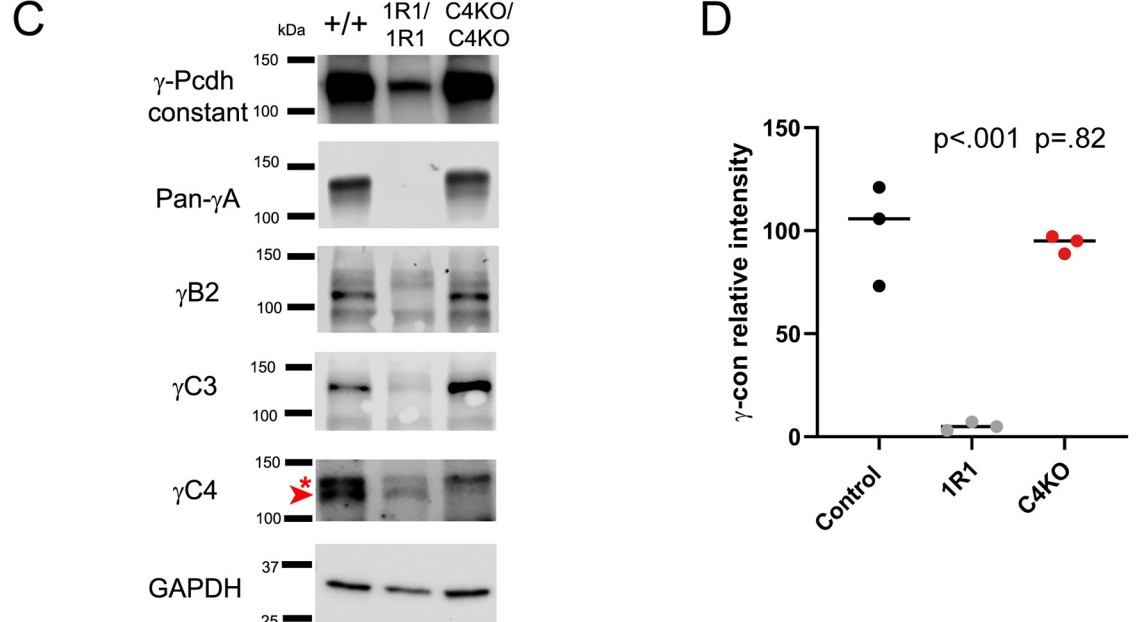

**Fig 7. Two further mouse alleles that indicate γC4 is the sole γ-Pcdh isoform necessary and sufficient for postnatal survival. A)** A schematic of the 1R1 allele illustrates a large deletion from the sgRNA guide site in exon A1 to that in exon C3, as well as a frame-shifting insertion in exon C5 (red upward triangle). Only exon C4 remained intact. All mutations were verified by whole-genome sequencing. **B)** C4KO mutants were made by targeting exon C4 only for CRISPR/Cas9 genome editing, resulting in a 13 base pair deletion (red downward triangle). **C)** Western blot analyses on P0 cortex confirmed the protein expression expected from sequence analysis. Particularly, γC4 was present in 1R1 mutants (arrowhead, specific lower band of expected size; asterisk indicates larger non-specific band) but absent in C4KO animals, while all other isoforms analyzed were absent in 1R1 but present in C4KO. **D)** Total γ-Pcdh protein levels were quantified using Li-Cor quantitative western blotting, normalizing to GAPDH. Values from three samples per genotype were compared to wild type controls using a Dennett's multiple comparison test with the indicated p-values.

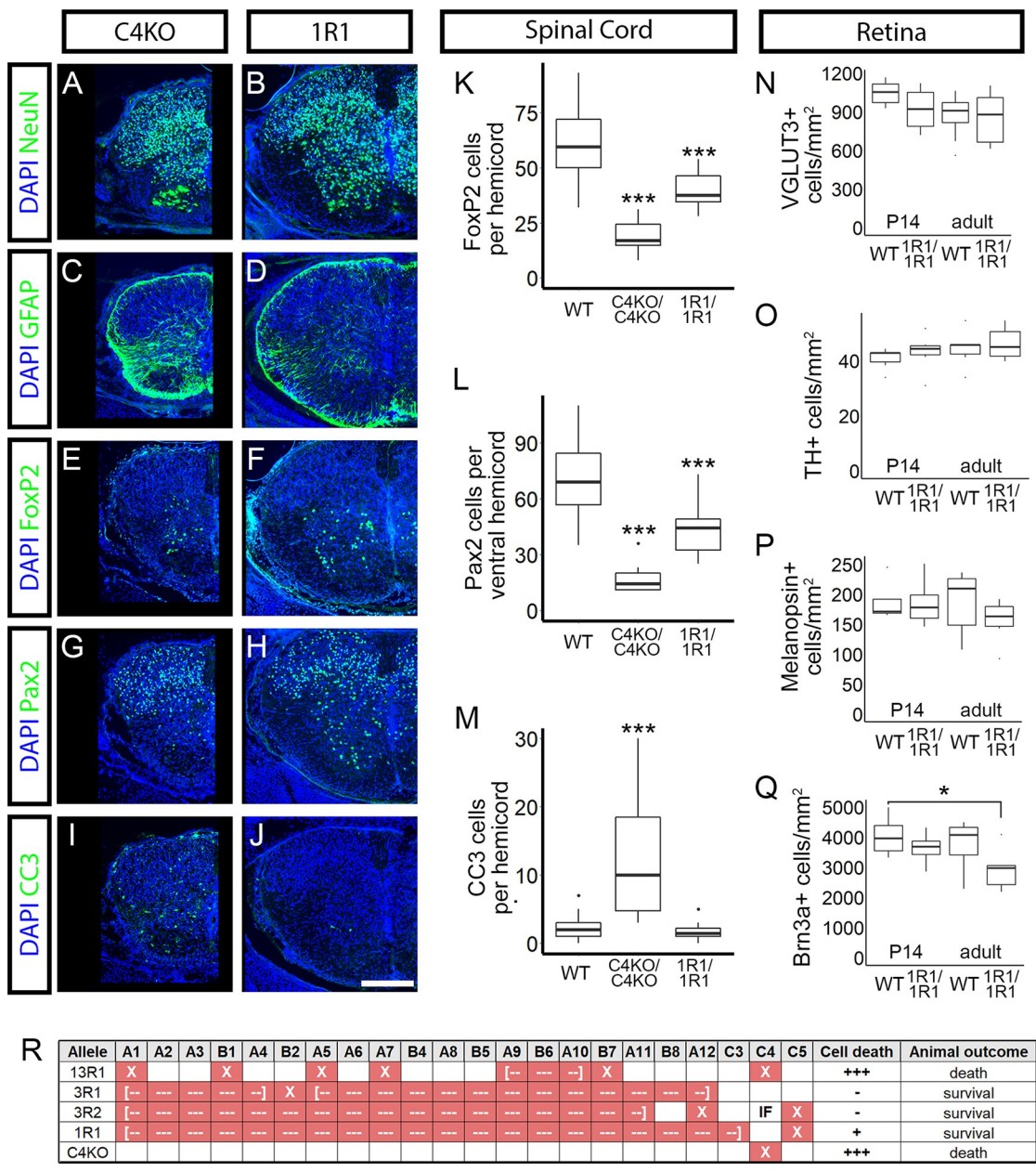

**Fig 8. The isoform γC4 is necessary for normal neuronal survival.** Cryosections from P0 spinal cords were immunostained for the indicated markers (green; DAPI counterstain for nuclei, blue) **A-B**) Immunostaing for NeuN revealed grossly smaller spinal cords and neuronal loss in C4KO mutants, but not 1R1 animals. **C-D**) This was accompanied by reactive gliosis in C4KO. **E-H**) Interneuron subtypes were analyzed as in Fig 4 (**E-F**, FoxP2 and **G-H**, Pax2). **K-L**) Ventral interneurons were drastically reduced in C4KO mutants; decreases in 1R1 mutants were statistically significant, but substantially more modest. **I-J**) Apoptosis was increased in C4KO animals, but not in 1R1 mutants, as demonstrated by CC3 immunolabeling (quantified in **M**). **N-Q**) Whole mount retinas from 1R1 mutants were assayed for neuronal loss at 2 weeks of age and in adult as in Fig 6. There were no reductions in (**N**) VGLUT3+ amacrine cells, (**O**) TH+ dopaminergic amacrine cells, or (**P**) Melanopsin+ ipRGCs. **Q**) At 2 weeks of age, Brn3b + RGCs were also not reduced, but their density was significantly lower in adult retinas. **R**) Summary of phenotypes in the analyzed strains. Mutations that disrupted *Pcdhgc4* resulted in high levels of cell death (+++) and postnatal lethality, while mutations that spared *Pcdhgc4* resulted in little to no cell death and animal survival. Scale bar is 200 μm. * = p < 0.05; ** = p < 0.01; *** = p < 0.001 by Dunnett (spinal cord) or Tukey (retina) test comparing the indicated genotype with wild type. Wild type values in K-M are represented from Fig 4. n = 36 hemicords total from 3 animals per genotype or n = 6 retinas per genotype. Box plots represent the median, first and third quartile, range, and outliers.

cell densities were calculated (in cells per mm$^2$) for four cell types: dopaminergic amacrine cells (TH+), VGLUT3+ amacrine cells, ipRGCs (Melanopsin+), and Brn3a+ RGCs (Fig 8O–8R). The means of the cell densities were compared across genotype and age with a two-way ANOVA followed by pairwise comparisons with a Tukey analysis. There was a significant genotype effect in Brn3a+ cell density (p = 0.034), and a significant pairwise difference between wild type animals at 2 weeks and 1R1 mutants in adult (p = 0.046). No other differences reached statistical significance. Therefore, neuronal survival was largely normal in 1R1 mutants, with any cell loss being very modest compared to mutants lacking γC4 (13R1/cRKO: Fig 5, Fig 6; *Pcdhg* nulls: [43]).

## *PCDHGC4* is constrained in humans

With the number of human genomes and exomes that have been sequenced, it is possible to test if variation occurs at the rate expected from random mutation. If predicted loss of function (LOF) mutations (e.g., frame-shifts and premature stop codon insertions) within a given gene are observed at lower rates than expected, it indicates that that gene is essential, and that loss of function is not tolerated (referred to as "constraint"). To ask if specific cPcdh isoforms are constrained in the human population, we queried the Genome Aggregation Database (gnomAD, Broad Institute), an aggregation of genomic variation across 141,456 human exomes and whole genomes [58]. For each cPcdh isoform, the ratio of observed to expected (o/e) LOF mutations was reported along with its 90% confidence interval (CI, vertical lines, Fig 9A and 9B). The suggested threshold for considering a gene under constraint is if the upper bound of the 90% CI falls below 0.35 (red line, Fig 9A and 9B) [58]. Within the *PCDHG* locus, only *PCDHGC4* met this criterion (o/e = 0.14, 90% CI = 0.07–0.31, Fig 9A). Amongst the *PCDHA* and *PCDHB* isoforms, only *PCDHAC2*, the specific isoform required in serotonergic neurons, was constrained (o/e = 0.16, 90% CI = 0.09–0.34). Thus, consistent with our analysis of the orthologous gene in our allelic series of mice, *PCDHGC4* is likely also essential in humans.

## Discussion

The 22 γ-Pcdhs comprise a diverse family of cadherin superfamily adhesion molecules with multiple distinct functions in distinct cell types. Extant data suggest three possible models for the role of isoform diversity: (1) a model of *diversity* where many isoforms are required for a given function; (2) a model of *redundancy* in which any single isoform (or small subset of isoforms) can serve a given function; and (3) a model of isoform *specificity* where one specific isoform (or small subset of isoforms) is strictly required for a given function. Here, we used a CRISPR/Cas9 strategy to reduce γ-Pcdh isoform diversity in an unbiased fashion, creating a new allelic series of mouse strains. In the course of analyzing several new *Pcdhg* alleles, we discovered that the control of neuronal survival and postnatal viability is best described by the third model. We found–surprisingly, given its inability to reach the cell surface and mediate homophilic adhesion without other cPcdh *cis*-interaction partners–that γC4 is the sole necessary and sufficient isoform among the γ-Pcdhs. We do not rule out the possibility that members of the α and β cluster could contribute to this function by acting as essential co-receptors or carriers. The new lines we report here will be instrumental for future studies delineating the importance of isoform diversity to other γ-Pcdh functions including self-avoidance and dendrite arborization.

### Generation of a new allelic series with reduced *Pcdhg* isoform diversity

Our strategy was to inject Cas9 mRNA along with 20 individual sgRNAs targeting each of the *Pcdhg* V exons simultaneously. By screening G1 offspring from many founders, we collected

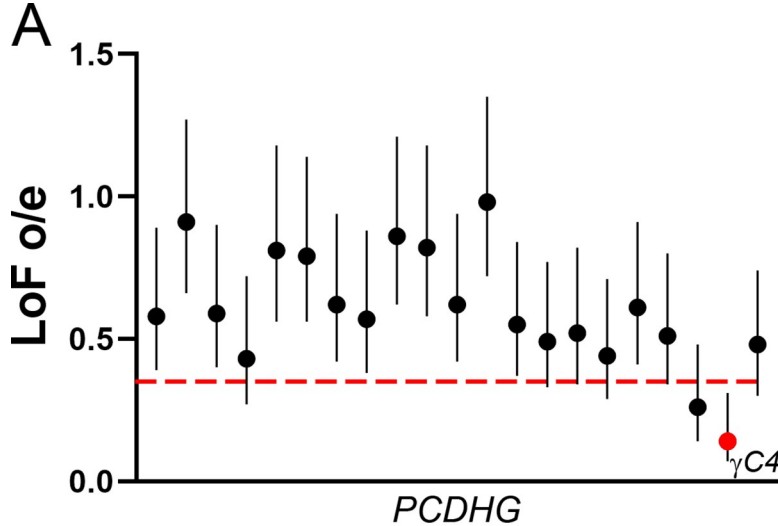

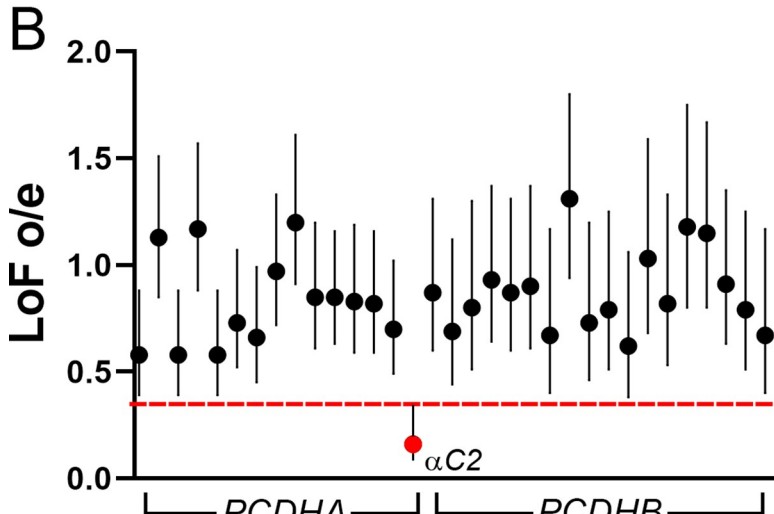

**Fig 9. *PCDHGC4* is constrained in humans.** The gnomAD database (https://gnomad.broadinstitute.org/; [58]) was queried for each cPcdh isoform to determine the extent to which loss-of-function (LoF) mutations were tolerated within the human population. **A)** The ratio of observed to expected (o/e) LoF mutations (y-axis) surrounded by its 90% CI is plotted for each *PCDHG* isoform (x-axis). *PCDHGC4* (red circle) is the only isoform in which the upper bound of the 90% CI falls below 0.35 (red line), indicating constraint. **B)** LoF o/e (y-axis) is plotted for each isoform from the *PCDHA* and *PCDHB* cluster (x-axis). *PCDHAC2* is the only constrained isoform (red circle).

an array of mutants with distinct V exon mutation patterns, ranging from a single isoform disrupted (21 intact) to 21 isoforms disrupted (1 intact), all of which were cryopreserved. Each isoform was disrupted in at least one mutant line (Table 1). Furthermore, there was evidence of double-stranded breaks at each sgRNA site identifiable as indels or junctions between guide sites. Not all guide sites were equally represented by mutations in the allelic series (e.g., γA1 was targeted in 16/26 new alleles, while γA11 was targeted in 3/26). This could reflect differences in the efficiencies of individual guides (only γC3 and γC4 sgRNA were tested individually) or differential accessibility of target sites. Our G1 screening was done by Illumina sequencing of a custom amplicon array. This technique was effective for identifying indels

within each individual amplicon (e.g., smaller indels at each guide site), but was not exhaustive in identifying rearrangements between guide sites, even when both sides of the junction were predicted to be recognized by the amplicon array primers. This could be due to the new DNA sequence: 1) being difficult to amplify within the array, and therefore not sequenced; 2) not being accurately aligned to the reference genome; or 3) not being identified as a new junction by the bioinformatic algorithm BreaKmer. Indeed, some junctions that were not recognized by BreaKmer were clearly identified by visual inspection of the sequence alignments within IGV as paired reads mapping to two different exons (S1 Table). Those identified in two different mutant lines– 3R1 and 1R1 –were confirmed by whole genome sequencing. It is thus likely that many of the cryopreserved lines not yet recovered and made homozygous harbor additional mutations to those described here from the initial analysis (Table 1, S1 Table). We focused on founders generated from the HI concentration of sgRNA (50ng). It is possible that founders from the MID or LO injection concentrations are more likely to harbor discrete indels and less elaborate rearrangements, but with lower total targeting efficiency. However, very few founders from the LO concentration were identified with any mutation in the initial screen, and alleles generated from the MID concentration still contained multiple rearrangements (e.g., *Pcdhg*$^{em33}$, S1 Table). We suggest that future studies targeting multiple sites within a relatively small region balance these concerns. We would, however, recommend methods other than amplicon sequencing to identify rearrangements between guide sites, such as targeted long-read sequencing [59].

One concern when using a pooled sgRNA approach such as ours is the compounding of potential off-target mutations. To assess this, we sequenced the top 95 predicted off-target sites in G1 offspring by amplicon sequencing. We did not identify indels attributable to Cas9 activity at any of these sites. While not exhaustive, this analysis is consistent with recent studies suggesting that off-target mutations from CRISPR/Cas9 are relatively rare in mouse model production[60–62]. To further confirm the specificity of our CRISPR/Cas9 targeting, we performed whole genome sequencing on homozygous mutants from four lines at least three generations after G1. Here, we were particularly concerned with the closely linked and highly similar protocadherins encoded by the two adjacent gene clusters (*Pcdha* and *Pcdhb*) as well as *Pcdh1* and *Pcdh12*, non-clustered protocadherins located within 450 kb of the *Pcdhg* cluster. There were no indels or rearrangements identified within any of these genes. We conclude that, at the resolution assayed here, off-target mutations do not contribute to our results. This conclusion is bolstered by the segregation of postnatal and neuronal survival phenotypes observed across multiple mouse lines: the two lines harboring frame-shifting mutations in *Pcdhgc4* (13R1 and C4KO, generated separately with distinct methodologies) died and exhibited similarly exacerbated neuronal apoptosis, while the two mouse lines in which *Pcdhgc4* was untouched (3R1) or harbored a small in-frame deletion (3R2) survived and exhibited normal neuronal survival. The fact that mice harboring an additional allele (1R1) in which *Pcdhgc4* is the only remaining functional isoform are viable and exhibit only mild neuronal survival alterations further supports the specificity of CRISPR targeting.

Isoform choice in the Pcdh gene clusters is regulated by CCCTC-binding factor (CTCF) and cohesin, which bind directly to a conserved sequence element (CSE) in each individual V exon promoter [63–65]. These proteins organize DNA loops, bringing the promoters of expressed isoforms into proximity with an enhancer region 3' to the constant exons [66]. CTCF/cohesin complex binding is restricted by promoter methylation laid down during embryogenesis by the methyltransferase Dnmt3b. Indeed, in the absence of *Dnmt3b*, many more cPcdh isoforms were expressed by each neuron [67]; conversely, in CTCF knockout forebrain neurons, expression of nearly all cPcdh genes was markedly reduced [68]. Here, we analyzed the isoform expression in cerebral cortex from four new *Pcdhg* mutant strains using

qPCR. In the *Pcdhg* cluster, indels or other genomic DNA junctions that allowed splicing from V exons to the constant exons resulted in expressed transcripts, while expression from isoforms deleted from the genome or inverted were undetectable, as expected (Fig 3, S7 Fig). The lone exception was the 3'-most isoform, γC5, which was significantly reduced in 1R1 mutants which harbored a single base pair insertion in the *Pcdhgc5* variable exon (S7 Fig). We also analyzed select isoforms from the *Pcdha* and *Pcdhb* clusters. Here, large deletions in the 5' end of the *Pcdhg* cluster resulted in significantly higher expression from the 3' *Pcdhb* isoforms (β11, β15, and β22, S3 Fig, S7 Fig). This was most pronounced in 1R1 mutants, which harbor the largest deletion in the *Pcdhg* cluster, but was not detected in 13R1 mutants, which have the smallest deletion of the mutants assayed. This is likely due to regulatory elements within DNaseI hypersensitive sites HS16-20, located 3' to the *Pcdhg* cluster, that are essential for expression from the *Pcdhb* cluster [69]. The movement of these elements closer to the *Pcdhb* cluster in the 1R1, 3R1, and 3R2 alleles is likely responsible for the increased expression observed. Consistent with this interpretation, it has previously been reported that moving *Pcdha* isoforms closer to regulatory elements located between the *Pcdha* and *Pcdhb* cluster (HS7 or HS5-1) by deleting intervening exons resulted in their increased expression [70]. Importantly, this increased expression of *Pcdhb* cluster genes cannot be responsible for the viability of the 1R1, 3R1, or 3R2 mouse lines: similar *Pcdhb* overexpression was reported previously in *Pcdhg*<sup>del/del</sup> animals, which nevertheless die at birth [34].

## *Pcdhgc4* is the essential isoform for neuronal survival and postnatal viability

Previous interrogations of *Pcdhg* isoform diversity either deleted two sets of three V exons (A1-A3 or C3-C5) [34] or drove overexpression of a single isoform from a transgene [40, 42, 44, 45]. Our new series of mutants, including both those initially analyzed and the many that remain cryopreserved, will allow for a finer dissection of the role of isoform diversity and the identification of potential isoform-specific functions. Chen et al. [34] showed that mice lacking γC3-C5 exhibited neonatal lethality and exacerbated spinal interneuron apoptosis similar to that of *Pcdhg* null mutants; in contrast, mice lacking A1-A3 were viable and outwardly normal. Our new results indicate that γC4 is, in fact, the sole essential isoform for neuronal survival and postnatal viability, indicating that loss of this isoform was responsible for the lethality observed in ΔC3-5 mice [34]. This is particularly surprising, as the γC4 isoform is peculiar in several ways. Cell aggregation experiments in the K562 suspension cell line indicate that γC4, uniquely among γ-Pcdhs, cannot mediate *trans* homophilic binding on its own, at least in heterologous cells *in vitro*. Like the α-Pcdhs, it requires interaction with carrier isoforms (either other γ- or β-Pcdhs) to reach the cell membrane [18], possibly because EC6 of γC4 inhibits surface delivery and the formation of *cis*-homodimers [17, 18]. This raises the novel possibility that the postnatal viability and largely (though not entirely) retained neuronal survival observed in 1R1 mutants, in which γC4 is the only functional γ-Pcdh isoform, may reflect functions for this protein in cellular compartments other than the plasma membrane. Alternatively, γC4 may reach the membrane (e.g., escorted by β-Pcdhs) and participate in homophilic interactions that trigger intracellular signaling pathways specific to the γC4 VCD, which remain to be identified. Precedence for this latter possibility comes from our recent finding that the VCD of γC3 can, uniquely amongst γ-Pcdhs, bind to and sequester Axin1 at the membrane, which leads to suppression of some components of the Wnt signaling pathway [49].

Thus, while the downstream mechanisms through which γ-Pcdhs promote neuronal survival remain unknown, our findings suggest that either protein interactions specifically mediated by γC4, or a unique localization for this isoform, will be involved. In 1R1 mutants, γC4

alone was not entirely sufficient for normal neuron number in the spinal cord or retina (Fig 8). This could be explained potentially by the significantly reduced expression levels of γC4 (and thus of total γ-Pcdh proteins) in these mutants (Fig 7C, S5 Fig), or it could indicate that other cPcdh isoforms contribute to survival of some neuronal subsets in a collaborative, if not strictly essential, manner. Consistent with this second possibility, cell death phenotypes observed in *Pcdhg* mutants were made more severe by additional disruption of the *Pcdha* and *Pcdhb* clusters, despite the fact that *Pcdha* or *Pcdhb* disruption did not increase cell death when *Pcdhg* remained intact [10, 48]. In any case, our genetic interrogation of the *Pcdhg* gene cluster clearly demonstrates that γC4 is the only γ-Pcdh isoform strictly necessary for neuronal survival and postnatal viability. Consistent with this, the orthologous *PCDHGC4* is the only *PCDHG* gene (and, along with *PCDHAC2*, one of only 2 clustered Pcdh genes overall) that is constrained in humans, thus indicating an essential role conserved throughout mammalian evolution that is of potential clinical relevance. In this respect, the critical next step will be to elucidate the unique protein-protein interactions and intracellular signaling pathways in which γC4 participates during the development of the nervous system.

## Materials and methods

### Ethics statement

All procedures using animals were performed in accordance with The Guide for the Care and Use of Laboratory Animals and were reviewed and approved by the Institutional Animal Care and Use Committee of The Jackson Laboratory (approval number 1026) and The Institution Animal Care and Use Committee of the University of Iowa (approval number 8011375), The Institution Animal Care and Use Committee of Wayne State University (approval number 18-06-0685). All animals were euthanized by CO2 asphyxiation, consistent with the recommendations of the American Veterinary Medical Association (AVMA) 2013 Guidelines on Euthanasia.

### Mouse strains

All animals were housed in the research animal facility either at The Jackson Laboratory, The University of Iowa, or Wayne State University under standard housing conditions with a 12 h/12 h light/ dark cycle and food and water ad libitum. All procedures using animals were performed in accordance with The Guide for the Care and Use of Laboratory Animals and were reviewed and approved by the Institutional Animal Care and Use Committee at each respective institution. All experiments included a mix of male and female animals. Previously described animals include *Pcdhg*[fcon3] [31] and *Pax6α-Cre* [57]. The newly generated reduced *Pcdhg* diversity mutants are cryopreserved at The Jackson Laboratory (Stock numbers listed in S3 Table).

### Generation of Pcdhg reduced diversity mutants

Guide RNA (sgRNA) sequences were designed to target the near 5' regions to the start codons of each variable exon (S4 Table). Guides were designed using the tool at crispr.mit.edu and were analyzed using RGEN tools to minimize off target sites and to maximize the likelihood of frameshifting mutations [71–73]. 20 total guides were synthesized (IDT), as exons B4 and B5 had a high level of 5' homology, as did exons B6 and B7. Guides were received lyophilized, resuspended in water, mixed in equal parts, then lyophilized again. This mixture was diluted and resuspended with *S. pyogenes* Cas9 mRNA to generate three guide concentrations: 50 ng/μl for each guide, 10 ng/μl for each guide, or 5 ng/μl for each guide (HI, MED, LO,

respectively). In each dilution, Cas9 mRNA was present at a concentration of 100 ng/μl. These mixtures were microinjected into C57BL/6J zygotes, which were subsequently implanted into pseudopregnant female C57BL/6J mice. 123 embryos injected with the HI concentration were transferred to 6 females, 152 at MED transferred to 8 females, and 123 at LO to 6 females.

The resulting live founders (16 HI, 34 MED, 50 LO) were screened by PCR and Sanger sequencing (primers in S5 Table). There were three iterations of screening. First, we chose 7 exons distributed across the locus for PCR amplification and Sanger sequencing in 100 live founders. Heterozygous and homozygous indels were identified by analyzing the sequence traces, and 15 founders were found to harbor some mutation. 50 of the 100 founders were from the LO injection condition. No mutations were found in the first round of screening in these animals, and they were not analyzed further. In the second round of screening, the 35 remaining founders were analyzed for indels at the other 15 exons. In the third round, they were screened for rearrangements between exons by PCR with single forward primers mixed with pooled reverse primers. PCR products detected above background level were purified and sequenced using the forward primer. Mice carrying any mutation detected in any round of screening were bred for one generation with C57BL/6J animals. The resulting G1 offspring were screened by amplicon sequencing. Males were prioritized when available for ease of sperm cryopreservation.

To generate $Pcdhg^{C4KO}$ mice, only sgRNA complimentary to exon C4 was microinjected (50 ng/μl, S4 Table) along with Cas9 mRNA (100 ng/μl). Of the resulting 16 live founders, 9 contained mosaic indels. 3 founders were crossed with C57BL/6J, one of which transmitted a 13 bp deletion. These mice were crossed to homozygosity, and all 22 variable exons were screened by PCR and Sanger sequencing to verify that only the *Pcdhgc4* exon was disrupted.

## Screening by amplicon sequencing

A TruSeq Custom Amplicon 1.5 assay was designed with Illumina DesignStudio. The amplicon length was 250 bp, 201 bp of which corresponded to target sequence. 153 targets were covered by 225 amplicons for a cumulative target length of 31,721 bp. Target regions were chosen to be centered at the sgRNA sites, the analogous sites in the variable exons in *Pcdha* and *Pcdhb*, and 95 predicted potential off-target sites. The target coordinates are listed in S1 File. Genomic DNA was extracted using the DNeasy 96 Blood & Tissue Kit (Qiagen) from 94 G1 animals and processed along with a C57BL/6J wild type control and a negative control. Libraries were constructed using the TruSeq Custom Amplicon (Illumina) Library prep kit. Briefly, custom probes were hybridized to the flanking region of interest in gDNA, DNA polymerase extended across the target region, and unique barcodes and sequencing primers are added by PCR. Libraries were pooled and sequenced on a MiSeq instrument from Illumina with paired-end reads, 150 bp long.

Sequences were matched to samples by barcode and aligned to the target regions (Base-Space, Illumina). Variants were called using Genome Analysis Toolkit (GATK) [52]. As CRISPR/Cas9 mediated NHEJ events result in small to medium sized indels, but not SNPs [74], we focused on indels that exceeded the following thresholds: QUAL $\geq$ 850; DP $\geq$ 90; QD $\geq$ 5. We also filtered out indels detected in the wild type C57BL/6J control, as these reflect discrepancies from the reference genome. Subsequent whole genome sequencing confirmed these mutations in the *Pcdhg* locus, but not potential off-target mutations expected to be linked (e.g., *Pcdhb7*). Furthermore, one mutation that did not exceed our threshold was detected by whole genome sequencing (A12 in $Pcdhg^{em35}$). Therefore, S1 Table summarizes mutations detected in the *Pcdhg* locus above and below the threshold, as indicated.

Because of the pooled amplicon reaction, we reasoned that large rearrangements between guide sites could be detected by amplicon sequencing if enough sequence remained on either

side of the junction. To find these junctions we used BreaKmer [53]. We were able to detect some of these junctions with BreaKmer, but subsequent whole genome sequencing revealed rearrangements missed by BreaKmer in each sequenced mouse line. In some cases, this could be because deletions extended beyond the amplicon site on one or both sides of the junction. However, many junctions were flanked by enough amplicon sequence to expect them to be detectible in this data set. Indeed, by visually analyzing the read alignments in the Integrative Genomics Viewer (IGV, Broad Institute) we were able to find some of these junctions when mating paired-end reads mapped to different exons (S1B Fig). We used this approach to identify additional junctions in the cryopreserved mouse lines as indicated in S1 Table.

## Whole genome sequencing

Genomic DNA was isolated from homozygous mutants using the DNA Extraction from Fresh Frozen Tissue protocol (10X Genomics). Briefly, nuclei were isolated from tissue, lysed using proteinase K, then DNA purified using magnetic beads. Linked-read whole genome libraries were constructed using the Genome Chip Kit v2 (10X Genomics). Briefly, high molecular weight DNA was partitioned into Gel Bead-In-EMulsions (GEMs) where unique barcoded primers were added to individual molecules of DNA. After the GEMs were dissolved, Illumina specific sequencing primers and barcodes were added by isothermal amplification, then library construction was completed via end repair, a-tailing, adapter ligation, and amplification. Libraries were sequenced on a HiSeq X (Illumina) by Novogene (Sacramenta, CA) with 150 bp paired-end reads. Sequences were aligned using the Long Ranger analysis pipeline from 10X Chromium and visualized in Loupe (10X Genomics) or IGV (Broad Institute). Paired-end reads were used to find NHEJ junctions between guide sites, while reads spanning the junctions were analyzed to uncover the specific sequences of these junctions.

## Data deposition

All Illumina sequencing data is available at the Sequence Read Archive (SRA) with the accession number PRJNA562238. Analysis outputs are at Mendeley with DOI: 10.17632/yhn7dpmv6v.1

## Quantitative RT-PCR

RNA was isolated from cerebral cortex of animals at P0 (13R1 mutants and littermate controls) or 2–12 weeks of age (other mutants and littermate controls) using Trizol reagent. Five μg of RNA per sample was used to make cDNA using Superscript III (Invitrogen) according to the manufacturers protocol. qPCR was performed in triplicate (technical replicates) using primers listed in S2 Table with SYBR Green PCR Master Mix. The relative abundance of each transcript was calculated using the ΔΔCt method, normalized to *GAPDH* and littermate controls. Briefly, cycle thresholds (Ct) were averaged across three technical replicates and the Ct of sample-matched *GAPDH* was subtracted (ΔCt). These ΔCt values were subtracted from the average ΔCt control value within litters (ΔΔCt), and the relative normalized expression calculated as 2^(ΔΔCt). The relative levels of each transcript in controls, 3R1, and 3R2 mutants were compared using an ANOVA with Tukey post-hoc tests. 13R1 and 1R1 mutants were compared with littermate controls for each transcript using a t-test.

## Western blotting

Whole brains at P0 or adult ages (3–8 months) were homogenized in RIPA buffer (0.1% SDS, 0.25% sodium deoxycholate, 1% NP-40, 150 mM NaCl, 50 mM Tris-HCl, pH 7.4, 5 mM NaF)

plus protease inhibitors (Roche Mini cOmplete) using a Dounce homogenizer and a Wheaton overhead stirrer. The lysate was centrifuged at 16,000 X g for 15 min at 4˚C to remove cell debris and proteins were quantified using a BCA assay kit (Pierce/Thermo Scientific). Forty μg of protein was resolved on Mini-PROTEAN TGX precast 7.5% SDS-PAGE gels (Bio-Rad) and proteins were transferred to a nitrocellulose membrane using the Bio-Rad Trans-Blot Turbo Transfer System. Membranes were blocked using 5% skim milk for 1 hour and incubated with the primary antibody in buffer (2.5% BSA in TBS-T (Tris buffered saline with 0.1% Tween-20) overnight at 4˚C. The following day, membranes were washed using TBS-T and incubated in HRP-conjugated secondary antibody for 1 hour. Membranes were washed, developed using the SuperSignal West Pico ECL reagents (Thermo Scientific), imaged using a Li-Cor Odyssey Fc Imaging system, and quantified using Li-Cor software. Levels in each lane were normalized to GAPDH.

## Immunofluorescence

Tissues were processed and stained as described previously [29, 31]. Briefly, neonatal spinal columns were removed and fixed by immersion in 4% paraformaldehyde (PFA) for four hours at 4˚ C, followed by extensive washing in PBS and cryopreservation in 30% sucrose. Eyes were enucleated and dissected to remove the cornea and lens, then fixed overnight by immersion in 4% PFA at 4˚ C followed by cryopreservation in 30% sucrose. Tissue was embedded in OCT (Sakura-Finetek) and sectioned with a cryostat onto positively charged Superfrost Plus slides (Fisher Scientific). After blocking in 2.5% bovine serum albumin with 0.1% Triton-X-100 in PBS, primary antibodies were incubated on the slides overnight at 4˚ C in a humidified chamber, followed by secondary antibodies for 1 hour at room temperature. Whole mount retinas were stained free floating in primary antibody diluted in blocking solution with 0.5% Triton-X-100 for 48–72 hours, and in secondary antibody for 24 hours. Sections were counter-stained with DAPI (4',6-diamidino-2-phenylindole), prior to mounting with Fluoro-Gel mounting media (Electron Microscopy Services #17985–11).

## Antibodies

Primary antibodies used included the following: Guinea pig anti-VGLUT3 (1:10,000, Millipore), sheep anti-tyrosine hydroxylase (1:500, Millipore), rabbit anti-melanopsin (1:10,000, ATS), mouse anti-Brn3a (1:500, Millipore), goat anti-choline acetyltransferase (1:500, Millipore), mouse anti-NeuN (1:500, Millipore), mouse anti-GFAP (1:500, Sigma), rabbit anti-FoxP2 (1:4000, Abcam), rabbit anti-Pax2 (1:200, Zymed), rabbit anti-cleaved caspase 3 (1:100, Cell Signaling Technologies), mouse anti-parvalbumin (1:500, Sigma), and mouse anti-GAPDH (1:500, Abcam). Mouse monoclonal antibodies against γ-Pcdh proteins used for western blots (1:500–1:1000) were generated by NeuroMab in collaboration with the Weiner laboratory [55] and obtained from Antibodies, Inc.: N159/5 (detecting an epitope in constant exon 1 or 2 and thus all 22 γ-Pcdh isoforms); N144/32 (detecting all γA subfamily isoforms); N148/30 (specific for γB2); N174B/27 (specific for γC3). A rabbit polyclonal antibody raised at Affinity BioReagents against the peptide sequence VAGEVNQRHFRVDLD (within EC1) from murine γC4 was also used for western blotting (1:1000). Secondary antibodies were conjugated with Alexa-488, -568, or -647 (1:500, Invitrogen) or HRP (1:1000–1:5000, Jackson Immunoresearch).

## Image quantification

Control and mutant spinal cords were imaged using epifluorescence at equivalent thoraco-lumbar locations using a Leica SPE TCS confocal microscope and captured with Leica

Application Suite software. From the resulting images, cell counts were performed using Cell Counter plugin in Fiji image analysis software [75]. For FoxP2 and CC3 analysis, all immuno-reactive cells were counted per hemicord from 12 hemicords per animal and at least 3 animals per genotype. For Pax2 analysis, all interneurons ventral to the central canal were quantified and compared. Values for wild type and mutant genotypes were compared using One-way ANOVA with Dunnett's multiple comparison test. Sample size was based on prior studies, as effective group sizes were known [31].

Whole mount retinas were imaged by confocal microscopy. In the resulting image stacks, cell density was measured using the Cell Counter plugin in Fiji image analysis software [75]. Values from at least two fields per retina, sampled from different regions midway between the center and periphery (technical replicates), were averaged. These averaged values from at least six retinas per genotype (biological replicates) were compared using an ANOVA with Tukey post-hoc tests. Sample size was based on prior studies, as effective group sizes were known. Retinas were analyzed at 2 weeks of age, with the exception of 1R1 mutants and littermate controls, which were analyzed at 2 weeks and adult (3–6 months of age). Here, values were compared using a two-way ANOVA with age and genotype as independent variables, followed by Tukey pairwise comparisons.

All p-values are listed in S3 File.

## gnomAD database analysis

For each cPcdh isoform in human, the gnomAD browser was queried (gnomAD. broadinstitute.org; [58]) and the ratio of observed to expected (o/e) loss of function variants was recorded along with its 90% confidence interval.

## Supporting information

**S1 Fig. Workflow for identifying mutations with 3R1 as an example. A**) PCR from genomic DNA in 3R1 homozygous mutants compared with wild type, using primer pairs spanning each sgRNA target site. PCR products were either absent or abnormal in size from more exons than predicted by the custom amplicon sequencing analysis. **B**) Amplicon sequencing reads from a heterozygous 1R1 mutant were aligned to the mouse genome and visualized in the Integrated Genome Viewer (IGV) with paired-end reads highlighted. In this example, multiple read pairs map with one end in exon A1 and the other in exon A4, indicating a junction between A1 and A4 that was not detected by BreaKmer. **C**) Linked-read whole genome sequencing was performed on DNA from a homozygous 3R1 mutant. The matrix represents the locations of mapped reads with common barcodes, indicating that they originated from common DNA fragments within a microfluidic droplet. The *Pcdhg* locus is presented on the X and Y axes, revealing three large deletions including the deletion resulting in the A1-A4 fusion found in **B**. **D**). Linked-read sequencing also revealed that *Anp32a* coding sequence incorporated into the Pcdhg locus, as reads mapping to its exons (but not introns) had barcodes in common with those mapping to *Pcdhg* (*Anp32a* is on the Y axis, *Pcdhg* on the X axis). (TIF)

**S2 Fig. Linked-read whole genome sequencing of two further *Pcdhg* mutant lines.** 10X Chromium linked-read sequencing results from (**A**) 13R1 and (**B**) 3R2 mutants demonstrate normal coverage through the *Pcdha* and *Pcdhb* clusters (upper panel), but coverage gaps where sequence was deleted in the *Pcdhg* locus (upper and lower panels). Short reads with the same barcode (i.e., from the same initial larger fragment) are connected on the matrix in the

lower panel. Actual read sequences were used to identify junctions.
(TIF)

**S3 Fig. Large deletions in *Pcdhg* increased mRNA expression from the 3' end of *Pcdhb*.**
Quantitative real-time PCR of cDNA reverse-transcribed from RNA isolated form cerebral
cortex from 13R1 mutants (red), 3R1 mutants (blue) and 3R2 mutants (green) demonstrated
no change in isoform expression from the *Pcdha* cluster genes analyzed. *Pcdhb* isoforms at the
3' end of the cluster were increased in mutants with large deletions in *Pcdhg* (i.e., in 3R1 and
3R2 mutants, but not in 13R1 mutants). * = p < 0.05; ** = p < 0.01 by Tukey post-hoc test
comparing the indicated genotype with wild type. n = 3–9 animals per genotype. Box plots represent the median, first and third quartile, range, and outliers.
(TIF)

**S4 Fig. Aggregation of Ia afferent axon terminals in mutants with interneuron apoptosis.**
Parvalbumin staining of proprioceptive Ia afferent axons in spinal cord sections from P0 **A**)
wild type, **B**) 13R1 homozygous mutants, and **C**) 3R1 homozygous mutants reveals axon terminal clumping in 13R1, but not 3R1 or wild type animals.
(TIF)

**S5 Fig. *Pcdhgc3* homozygous mutants survive at expected ratios. A**) CRISPR/Cas9 targeting
of *Pcdhgc3* only resulted in a 13 bp deletion in the *Pcdhg*^C3KO^ allele (referred to as C3KO here).
**B**) Tail genotyping PCR spanning the deletion was used to identify wild type, heterozygous,
and homozygous mutants. **C**) Homozygous C3KO mutants survive in numbers not significantly different from the expected Mendelian ratio (n = 118 offspring from 18 litters).
(TIF)

**S6 Fig. Confirmation of 1R1 and C4KO mutations. A**) 10X Chromium linked-read sequencing results from 1R1 homozygous mutants demonstrates normal coverage through the *Pcdha*
and *Pcdhb* clusters (upper panel), but a large gap between *Pcdhga1* and *Pcdhgc3* (upper and
lower panels). Short reads with the same barcode (i.e., from the same initial larger fragment)
are connected on the matrix in the lower panel. Actual read sequences were used to identify
the junction. **B**) Sanger sequencing was performed on PCR from genomic DNA from C4KO
homozygous mutants. A frame-shifting 13 bp deletion was identified at the guide site in
*Pcdhgc4*, but no mutations were found in any of the other isoforms (*Pcdhgc3* and *Pcdhgc5* are
shown here, all other isoforms were also sequenced).
(TIF)

**S7 Fig. Quantitative RT-PCR from 1R1 mutants. A**) Quantitative real-time PCR of cDNA
reverse-transcribed from RNA isolated form cerebral cortex from 1R1 mutants (gray) compared to control (white) demonstrated no change in isoform expression from the *Pcdha* cluster. Expression of *Pcdhb* isoforms at the 3' end of the cluster was increased in 1R1 mutants,
consistent with the effect from large deletions in 3R1 and 3R2 mutants. **B**) Expression of the
*Pcdhg* cluster reflected genomic mutations, including expression from the γA1-γC3 fusion.
γC5 isoform expression was significantly reduced (disrupted by 1 base pair insertion), and
total γ-constant expression was reduced by half. * = p < 0.05; ** = p < 0.01; *** = p < 0.001 by
student's t-test. n = 3 animals per genotype. Box plots represent the median, first and third
quartile, range, and outliers.
(TIF)

**S8 Fig. γC4 is necessary and sufficient among γ-Pcdh isoforms for normal Ia afferent terminal targeting. A**) Parvalbumin staining in spinal cord sections from P0 C4KO mutants
revealed aggregation of proprioceptive Ia afferent axons comparable to that observed in null

animals or in 13R1. **B**) Conversely, 1R1 homozygous mutants exhibited normal terminal morphology similar to wild type, 3R1, and 3R2 pups.
(TIF)

**S1 Table. Methods used to identify mutations.** Mutations were identified by analyzing sequence results from the custom amplicon using GATK (blue = above threshold or purple = below threshold), Breakmer (orange), or visual inspection of aligned reads (yellow), or by analyzing whole genome sequencing (green). Numbers indicate the size of insertion (positive numbers) or deletion (negative numbers). Dashed lines indicate large scale rearrangements or deletions between the indicated exons.
(PDF)

**S2 Table. Primers used for quantitative RT-PCR.**
(PDF)

**S3 Table. Summary of new alleles created here.**
(PDF)

**S4 Table. sgRNA sequences.**
(PDF)

**S5 Table. Primers used to amplify target regions of *Pcdhg* variable exons.**
(PDF)

**S1 File. Genomic coordinates of custom amplicon sequencing targets.**
(CSV)

**S2 File. Mutation details.** Tab 1 includes GATK calls for mutations detected by custom amplicon sequencing. Tab 2 includes BreaKmer calls. Tab 3 includes primary protein sequence for all mutant isoforms from the lines analyzed in detail.
(XLSX)

**S3 File. p-values from all analyses.**
(XLSX)

## Acknowledgments

We would like to thank the scientific services at the Jackson Laboratory for assistance throughout this project, including Genetic Engineering Technologies, Microinjection, and Reproductive Sciences services for the production and preservation of new mutants, the Genome Technologies service for sequencing, and the Bioinformatics service for data analysis. We would also like to thank Kate Miers for assistance with the mouse colony.

## Author Contributions

**Conceptualization:** Andrew M. Garrett, Joshua A. Weiner, Robert W. Burgess.

**Data curation:** Andrew M. Garrett, Preeti Bais.

**Funding acquisition:** Joshua A. Weiner, Robert W. Burgess.

**Investigation:** Andrew M. Garrett, Peter J. Bosch, David M. Steffen, Leah C. Fuller, Charles G. Marcucci, Alexis A. Koch, Preeti Bais.

**Methodology:** Andrew M. Garrett, Peter J. Bosch, David M. Steffen, Leah C. Fuller, Charles G. Marcucci, Alexis A. Koch, Preeti Bais.

**Project administration:** Joshua A. Weiner, Robert W. Burgess.

**Supervision:** Joshua A. Weiner, Robert W. Burgess.

**Writing – original draft:** Andrew M. Garrett, Joshua A. Weiner, Robert W. Burgess.

**Writing – review & editing:** Andrew M. Garrett, Peter J. Bosch, David M. Steffen, Leah C. Fuller, Charles G. Marcucci, Alexis A. Koch, Preeti Bais, Joshua A. Weiner, Robert W. Burgess.

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
