## [Decision Letter · Decision Letter 0]

15 Nov 2019

Dear Dr Burgess,

Thank you very much for submitting your Research Article entitled 'CRISPR/Cas9 interrogation of the mouse Pcdhg gene cluster reveals a crucial isoform-specific role for Pcdhgc4' to PLOS Genetics. Your manuscript was fully evaluated at the editorial level and by three independent peer reviewers. As you can see, the three reviewers used very positive terms and found your findings interesting. No major technical issues were raised, but a number of suggestions were made that were referred to as minor points. Please, carefully read these suggestions and address them properly, since I think that they can help to improve the quality of your manuscript. We will wait until you address these suggestions before we make a final decision. 

[LINK]

Yours sincerely,

Ivan Garcia-Bassets

Guest Editor

PLOS Genetics

Gregory Barsh

Editor-in-Chief

PLOS Genetics

Reviewer's Responses to Questions

**Comments to the Authors:**

Reviewer #1: This is a big, excellent paper that reports two important sets of results. One part introduces a large (n=26) panel of mutant lines with isoform-specific deletions in the gamma protocadherin (pcdhg) gene cluster generated by a novel CRISPR-based method. The second part uses some of these lines, along with others, to demonstrate a role for one particular isoform, gC4. This is particularly noteworthy because there are very few cases in which a single isoform has a role that is not shared by other isoforms.

The work is extremely complete, well done and well described. I have no technical concerns or requests for additional data; my comments are only stylistic.

Line 23: C4 as “the only…that cannot independently engage…” This is work from others and is not, in my view decisive, as it used heterologous cells. I don’t know of any evidence that this pattern holds in neurons so unless you do, I don’t think making this point gains you anything. I therefore suggest de-emphasizing it here, and adding appropriate caveats when you return to it in the discussion.

Lines 34-54: You make it sound like the Pcdhs are Dscam wannabes. The parallels are striking, but it would be more straightforward to start with Pcdhs, then make the comparison.

Line 95: extant?

Lines 109-110: Please indicate whether the role of aC2 is because it is the only one that that can do the trick, or just the only one expressed.

Lines 112-123: It would help the reader to point out here that the paper has two somewhat separate parts, one on generation of many lines and the other on gC4. As it stands, you sort of short change the first part, which is quite elegant in its own right.

Line 291: This ends a very long section showing that 13R1 has a severe phenotype, more or less equivalent to the Pcdhg null. This is of interest but is not directly relevant to what came before (the 26 lines), and it is relevant to what follows in that C4 is deleted but C3 and C5 are retained. With this in mind, it might help the reader to shorten this section and provide a final few sentences emphasizing the relevance to the second part.

Line 316: Why not add the targeted C4 deletion and perhaps fCON to Table so it will be complete. They could be separated from the em series by a space or a heavy line to indicate that they were generated by a distinct strategy.

Line 350: Of course it would be interesting to know whether self-avoidance is disrupted in 1R1. This is far beyond the scope of this paper, but perhaps worth mentioning as an issue to be investigated in the future. (Another point for future attention, that you could but need not mention in the discussion is the use of the C4 only line to investigate C4 trafficking in neurons.)

Line 279: As in the introduction, the two important parts of the paper (lines and C4) are run together here. I think it would help highlight the strengths of the paper to discuss them under separate subheadings.

Reviewer #2: This article is an elegant and ambitious study that applied CRISPR mutagenesis to functionally interrogate the roles of clustered Protocadherin diversity in nervous system development. The cPcdhs are encoded by ~60 genes organized into 3 clusters and regulate neuronal survival and multiple aspects of neuronal patterning. The complexity of the locus has made conventional genetic studies very challenging. Here, the authors applied a CRISPR strategy to mutagenize a complex locus in mice. They generated several mouse alleles, followed with multiple sequencing platforms to characterize in detail the resulting alterations, and then performed phenotypic analyses to assess the functional requirements for the Pcdhs genes. The findings are well described and the conclusions are supported by these findings. The results are detailed and I especially appreciate that they link the resulting genomic mutations to changes in expression of targeted and non-targeted Pcdh genes, to phenotype.

This manuscript provide significant findings that advance the understanding of Pcdh functions. Importantly, they highlighted 3 potential models for the functional roles of this diverse gene family. Through their approach and analyses, they provide support for one model by demonstrating that a particular member of this large cluster has a specific, essential requirement for neuronal survival. It also informs future studies on dissociating the Pcdhg-dependent mechanisms that regulate survival, versus those that regulate neuronal patterning.

Secondly, this study is also significant to the broader field of functional genomics in mice, also it provides an elegant framework to manipulate and study the functional diversification of clustered genes. I expect this study will appeal to a broad readership.

Reviewer #3: In this study, Garrett et al. report the generation of a collection of 26 knockout mouse lines that lack the expression of one or different subsets of Pcdhg isoforms. After deeper examination of five of them, the authors reach three major conclusions: (1) that Pcdhgc4 plays a critical role in neuronal survival; [2] that this role does not require the diversity of the rest of the cluster; and, [3] that this role would be played through a mechanism that is not the expected based on what we currently know about gC4.

Overall, this is an experimentally correct study, thoroughly conducted, and the findings are of relevance. I only suggest a few minor points that, in my opinion, could further improve the manuscript.

Minor points:

(1) Lines 71-73: Although the stated conclusion that “many” neurons express gC4 is correct, I think that it does not provide an accurate description of what the cited study shows, which is that perhaps half of the neurons do not express gC4.

(2) Lines 128: in their core experiment with pooled gRNAs (50ng), the authors identified a large number of genomic rearrangements and alterations that complicated the subsequent characterization and, perhaps, caused some confounding effects; it is unclear. There are two points related with this comment:

- In the Methods section (or the Discussion), it would be useful to find a discussion about the caveats of the experimental design and recommend alternatives (in case anyone wants to do something similar in the future). For instance, would the authors have preferred using the 5/10ng concentrations to generate the mouse lines after what they encountered with the 50ng concentration? The 5/10ng conditions, although less efficient, have likely resulted in fewer deletions and inversions and more single, frameshift mutations. Would not have been ideal to work with conditions close to or lower than one gRNA per cell (as 5 and 10ng likely represent)? Would the authors recommend verification using long-read instead of short-read sequencing approaches (such as nanopore) to more easily resolve rearrangements and inversions?

- I suggest adding a supplementary table describing all the mutations at the nucleotide level, including their distance to ATG and distance to the new, predicted stop codon, the length of the predicted in-frame portion until the mutation, and other. I missed this information multiple times during the revision. I was trying to get a sense whether truncated proteins could, potentially, cause confounding effects. It is relevant to interpret the data, especially since underlying mechanisms of the gC4 actions are still unknown; so, the more we know, the better.

(3) Line 134: (Style). Since the figure tries to be illustrative, I would suggest showing one or several examples of frameshift mutations and their distance to the gRNA (in Figure 1C). In Figure 1D, I would provide relevant info (zygote number? Number of founders? Concentrations? Mouse genotype? How many cases passed each step?).

(4) Looking at Table 1, it is intriguing that gA1, gA3, gA5, and gA12 are targeted 16, 11, 9, and 8 times, respectively; whereas, gA4, hA6, gB2, gA8, and gA9 are targeted once or twice. I would appreciate a comment about this bias and number/statistics, if possible. I envision two scenarios: differences in gRNA sequence efficiencies, or differences in accessibility of the targeted regions. Did the authors tested their gRNAs individually first to have a sense of this? If they do not, I would make a comment about the possibility that some gRNAs did not work properly.

(5) Figure 3: Is fold change the best way to represent these data? (I do not understand a fold change of 0, or I did not find information about the reference used to determine 1). Would be a log-scale plot more appropriate to fairly show >1-fold and <1-fold changes? In some cases, the statistics seem odd (how can some significances be (*), (**), and (***) when the RNA seems not to be detected in the three cases? How can they be anything different than (***) if the levels detected are below detection?). The title of the figure would not be correct, expression does not reflect DNA mutations. Which housekeeping genes were used as reference? Western blots show high variability. For instance, constant levels are very different between WT mice. This variability is clear in two WT condition in Fig.3C, but not in the third WT condition. I am worried that this variability may affect the analysis of the mutant lines. Finally, I recommend farther separating the ‘GAPDH’ label from the arrow on the left panel, since the arrow seems indicating the position of the GAPDH band.

(6) One of the most interesting observations, in my opinion, is shown in Fig. 4E. It is a robust increase of GFAP+ astrocytic cells in the 13R1 line that can also be observed in the gC4KO line in Fig. 8C. The authors do not make any comment about this (or I did not find it). Is this another feature that can be associated with gC4? One interesting fact about gC4 expression (in the brain) is that it is largely anti-correlated with the emergence of the astrocytic population. I understand that this is completely beyond the scope of this paper, but it could be that, at the cell surface, gC4 inhibits glial differentiation around gC4-expressing neurons (external effect; involved in cell diversity), or that in the absence of gC4, neural progenitor cells chose a glial path (internal effect; involved in terminal differentiation). I would appreciate some discussion about these exciting observations.

(7) Line 229: In Fig. 3A, the authors concluded, “Total locus expression was significantly reduced in 3R2 homozygous mutants, but in both 3R1 and 13R1 mutant cortex, expression levels were indistinguishable from controls.” Even though the differences do not reach significance when comparing 3R1 versus control, there are obvious differences. Did the authors perform a power calculation to determine whether it is only a matter of numbers? In fact, the difference is also obvious in the Western blot shown in Fig. 3A.

**Have all data underlying the figures and results presented in the manuscript been provided?**

Reviewer #1: Yes

Reviewer #2: Yes

Reviewer #3: Yes

PLOS authors have the option to publish the peer review history of their article (what does this mean?). If published, this will include your full peer review and any attached files.

Reviewer #1: No

Reviewer #2: No

Reviewer #3: No

---

## [Editor Report · Decision Letter 1]

5 Dec 2019

Dear Dr Burgess,

We are pleased to inform you that your manuscript entitled "CRISPR/Cas9 interrogation of the mouse Pcdhg gene cluster reveals a crucial isoform-specific role for Pcdhgc4" has been editorially accepted for publication in PLOS Genetics. Congratulations!

Yours sincerely,

Ivan Garcia-Bassets

Guest Editor

PLOS Genetics

Gregory Barsh

Editor-in-Chief

PLOS Genetics

Comments from the reviewers (if applicable):

**Data Deposition**

http://datadryad.org/submit?journalID=pgenetics&manu=PGENETICS-D-19-01757R1

Press Queries

---

## [Editor Report · Acceptance letter]

20 Dec 2019

PGENETICS-D-19-01757R1 

CRISPR/Cas9 interrogation of the mouse *Pcdhg* gene cluster reveals a crucial isoform-specific role for *Pcdhgc4*  

Dear Dr Burgess, 

We are pleased to inform you that your manuscript entitled "CRISPR/Cas9 interrogation of the mouse *Pcdhg* gene cluster reveals a crucial isoform-specific role for *Pcdhgc4*  " has been formally accepted for publication in PLOS Genetics! Your manuscript is now with our production department and you will be notified of the publication date in due course.

With kind regards,

Matt Lyles

PLOS Genetics

On behalf of:
